# Defining neutralization and allostery by antibodies against COVID-19 variants

Nikhil Kumar Tulsian [1,2,12] ✉, Raghuvamsi Venkata Palur[1,3,12], Xinlei Qian[4,12], Yue Gu[4,5], Bhuvaneshwari D/O Shunmuganathan[4,5], Firdaus Samsudin[3], Yee Hwa Wong [6,7], Jianqing Lin [6,7], Kiren Purushotorman [4,5], Mary McQueen Kozma[4], Bei Wang [8], Julien Lescar [6,7], Cheng-I Wang [8], Ravindra Kumar Gupta[5,9,10], Peter John Bond [1,3] ✉ & Paul Anthony MacAry [5,11] ✉

The changing landscape of SARS-CoV-2 Spike protein is linked to the emergence of variants, immune-escape and reduced efficacy of the existing repertoire of anti-viral antibodies. The functional activity of neutralizing antibodies is linked to their quaternary changes occurring as a result of antibody-Spike trimer interactions. Here, we reveal the conformational dynamics and allosteric perturbations linked to binding of novel human antibodies and the viral Spike protein. We identified epitope hotspots, and associated changes in Spike dynamics that distinguish weak, moderate and strong neutralizing antibodies. We show the impact of mutations in Wuhan-Hu-1, Delta, and Omicron variants on differences in the antibody-induced conformational changes in Spike and illustrate how these render certain antibodies ineffective. Antibodies with similar binding affinities may induce destabilizing or stabilizing allosteric effects on Spike, with implications for neutralization efficacy. Our results provide mechanistic insights into the functional modes and synergistic behavior of human antibodies against COVID-19 and may assist in designing effective antiviral strategies.

A comprehensive understanding of antibody-mediated neutralization of SARS-CoV-2 is critical for development and evaluation of prophylaxes (vaccines) and antibody-mediated therapies for COVID-19[1,2]. The principal target for development of effective antibody-based antiviral approaches is the viral Spike glycoprotein[3–5]. In the Spike trimer, the three receptor-binding domains (RBDs) exist in an equilibrium of "up" or "down" positions[4,6]. In the up-position, the residues that interact with the human angiotensin converting enzyme-2 (ACE2) receptor become accessible for binding[7]. This induces conformational changes across the pre-fusion state of Spike trimer and may promote its cleavage by host proteases at the S1/S2 cleavage site, forming a post-fusion state that mediates entry into

[1]Department of Biological Sciences, National University of Singapore, Singapore 117543, Singapore. [2]Department of Biochemistry, National University of Singapore, Singapore 117546, Singapore. [3]Bioinformatics Institute, Agency for Science, Technology, and Research (A*STAR), Singapore 138761, Singapore. [4]Antibody Engineering Programme, Life Sciences Institute, National University of Singapore, Singapore 117546, Singapore. [5]Department of Microbiology and Immunology, Yong Loo Lin School of Medicine, National University of Singapore, Singapore 117546, Singapore. [6]School of Biological Sciences, Nanyang Technological University, Singapore 637551, Singapore. [7]NTU Institute of Structural Biology, Experimental Medicine Building, Singapore 636921, Singapore. [8]Singapore Immunology Network, Agency for Science, Technology and Research (A*STAR), Singapore 138648, Singapore. [9]Cambridge Institute of Therapeutic Immunology & Infectious Disease (CITIID), Cambridge, UK. [10]Department of Medicine, University of Cambridge, Cambridge, UK. [11]Life Sciences Institute, National University of Singapore, Singapore 117546, Singapore. [12]These authors contributed equally: Nikhil Kumar Tulsian, Raghuvamsi Venkata Palur, Xinlei Qian. ✉e-mail: nikhilkt.science@gmail.com; peterjb@bii.a-star.edu.sg; micpam@nus.edu.sg

host cells[6,8,9]. To curb the ongoing COVID-19 pandemic, multifaceted strategies such as mRNA based vaccines[10,11], small-molecule inhibitors[12–14], adenovirus-based vaccines[15], and antibody-based therapeutics have been deployed[16–18]. Neutralizing antibody-mediated therapies remain an effective antiviral strategy, as these can be rapidly targeted and/or tested against emerging variants and may also be useful for any future coronavirus-based pandemics[19]. Multiple neutralizing antibodies have been developed for COVID-19 that target either the Spike N-terminal domain (NTD) or RBD[20–22] (Fig. 1a), which interfere with interactions between the virus and the host receptors[23,24].

Since the beginning of the pandemic, the SARS-CoV-2 virus has undergone significant evolution, likely as a result of chronic infection within individual hosts[25] and immune pressure. Analysis of variants has identified mutations in various domains of the Spike protein which alter viral infectivity[26–28] and cell tropism[29]. In parallel with the emergence of new variants, knowledge of variant-specific conformational changes in the Spike protein has accelerated. For example, the NTD can modulate the efficiency of cleavage at the S1/S2 site and thereby impact of the cofactor TMPRSS2[29], or can be influenced by binding of the antibody 4A8[30], concomitant to the allosteric changes in Spike following ACE2 binding[31]. Amongst the various neutralizing antibodies

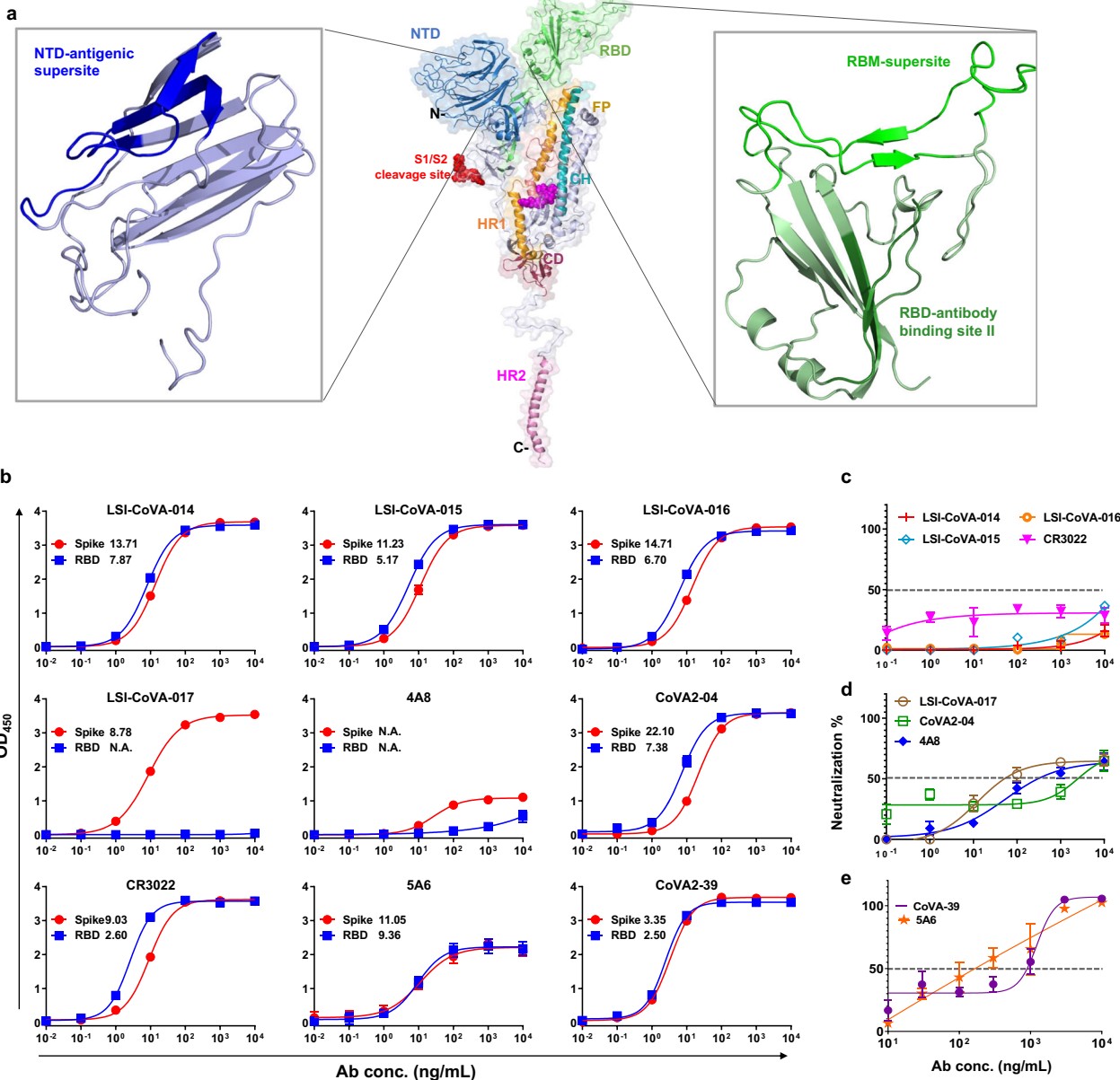

**Fig. 1 | Binding profiles and neutralization activities for human antibodies against Spike and RBD proteins. a** Illustration showing different functional domains of the SARS-CoV-2 Spike protein mapped onto a monomer. The antibody binding sites across the NTD (left, inset) and RBD (right, inset) are highlighted. **b** Antibody binding activity to Spike Hexapro (Wuhan-Hu-1) purified from mammalian cell culture and isolated RBD (Wuhan-Hu-1). Nine antibodies at varying concentrations from 10 pg/mL–10 μg/mL were tested for binding to SARS-CoV-2 Spike (red plots) and MBP-RBD (blue plots) by ELISA, and the $EC_{50}$ values ($n = 3$ independent experiments) are indicated. Data is represented as mean ± error bars (SEM). **c–e** Antibodies at varying concentrations (100 ng/mL–10 μg/mL) were incubated with a pseudotyped-virus lentiviral construct expressing the Spike protein, tagged with luciferase, followed by infection of CHO-ACE2 cells. The chemiluminescence-based luciferase assay readouts were then plotted and presented as percentage neutralization–grouped into **c** weak-, **d** moderate- and **e** strong-neutralizing antibodies based upon comparisons with WHO reference standards. Average values ($n = 3$ independent experiments) and standard deviations (mean ± SEM) are shown. Source data is provided as Source data file.

initially developed for the parent Wuhan-Hu-1 strain, only a minor fraction of the antibodies have been found to bind to or neutralize various variants[32], due to mutations at the antigenic supersites conferring immune escape[1]. Currently, there is a paucity of molecular level detail on the conformational changes induced by antibody binding.

In this study, we have characterized a group of novel human antibodies principally derived from convalescent blood samples and describe the nature and dynamics of their interactions with wild-type Spike protein and its variants. Using hydrogen-deuterium exchange mass spectrometry (HDXMS), in vitro assays, and molecular dynamics (MD) simulations, we have mapped the interaction interfaces of Spike and its variants with full-length antibodies (IgGs). Characterization of the effects of antibody binding to Wuhan-Hu-1, Delta (δ), and Omicron (o) variants of Spike uncovered mechanistic insights into the varying efficacy and potency among antibodies. This study provides a structure-based rationale for using antibody cocktails as an effective mode of neutralization of both former and future SARS-CoV-2 variants.

## Results

### Varying binding affinities and neutralization efficacies of Spike by antibodies

We performed biophysical characterization of nine human monoclonal IgG antibodies ('HuMAbs'), namely LSI-CoVA-014, LSI-CoVA-015, LSI-CoVA-016, LSI-CoVA-017 which were discovered in this study, along with 4A8[33], 5A6[34], CR3022[35], CoVA-02[36], and CoVA-39[36]. These monoclonal antibodies were principally discovered from convalescent patients of COVID-19 with the exception of CR3022 that was derived from a SARS-CoV-1 patient[35], and 5A6 which was derived from a naïve human phage-FAB library. The binding activity of each antibody to SARS-CoV-2 Spike trimer ('Spike') and isolated RBD ('RBD') from the Wuhan-Hu-1 strain was determined using Quartz Crystal Microbalance (QCM) and enzyme-linked immunosorbent assay (ELISA). As indicated by the half-maximal effective concentration ($EC_{50}$) values, seven of the nine antibodies bound strongly to both Spike trimer and RBD (Fig. 1b). Antibody LSI-CoVA-017 binds strongly to Spike but not RBD, suggestive of an epitope outside RBD. Antibody 4A8 binds weakly to Spike and showed negligible binding to RBD, consistent with previous studies that show 4A8 binds NTD[33]. Next, we determined the binding kinetics of these HuMAbs against Spike and observed high affinity binding with slow off-rates (Supplementary Fig. 1 and Supplementary Table 1). The affinity constants ($K_D$) were in the sub-nM range, with LSI-CoVA-017 being the lowest (0.088 nM). The association-dissociation kinetics clearly indicate stable binding of the HuMAbs to the Spike trimer.

We next investigated their neutralization efficacies using a pseudotyped virus neutralization test (PVNT). A neutralization capacity of >50% was considered significant in accordance with WHO standards. Correspondingly, we observed differential levels of neutralization, wherein LSI-CoVA-014, LSI-CoVA-015, LSI-CoVA-016, and CR3022 showed less than 50% efficacy. On the other hand, 4A8, LSI-CoVA-017, and CoVA2-04 showed significant neutralization capacities, while the highest neutralization was observed for CoVA2-39 and 5A6. On this basis, the antibodies were classified as (i) weak, (ii) moderate, and (iii) strong neutralizing HuMAbs (Fig. 1c–e).

### Weak neutralizing antibodies bind at a cryptic site on RBD and destabilize Spike trimer

The epitopes of the HuMAbs were mapped by comparative HDXMS analysis of complexes with Spike and RBD. We observed extensive protection against deuterium exchange across peptides spanning RBD of Spike and isolated RBD, in the presence of LSI-CoVA-014, LSI-CoVA-015, LSI-CoVA-016. This indicates binding to either the receptor-binding motif (RBM) or at a site distal to RBM (Fig. 2a–f). Overlapping peptides covering residues 361–395 showed large-scale protection against deuterium exchange in both Spike (Fig. 2h, i) and RBD

complexes with LSI-CoVA-014, LSI-CoVA-015, and LSI-CoVA-016, indicating that these three antibodies bind RBD at a site distal to the RBM site. In the trimeric Spike, the region spanning residues 361–395 becomes accessible only when the RBD adopts an up-position. These changes indicate that the epitope sites identified for LSI-CoVA-014, LSI-CoVA-015, and LSI-CoVA-016 are similar to the site observed for the CR3022 antibody (Supplementary Fig. 2a), previously characterized as a "cryptic" site binder[37]. Alongside antibodies discovered in the current study, we mapped the binding interfaces of known RBD-binding IgGs— CoVA2-04, CoVA2-39, and CR3022 (Supplementary Fig. 2). The binding hotspots mapped using HDXMS agree with reported cryo-EM structures[38]. Consistent with our expectations, increased deuterium exchange was observed across the peptides spanning the RBM/ACE2 binding site upon binding of either of these three antibodies (Fig. 2a–f and Supplementary Fig. 2a). This correlates to increased conformational dynamics at the ACE2 binding site, suggesting that binding of LSI-CoVA-014, LSI-CoVA-015, or LSI-CoVA-016 locks the RBD in the up-position resulting in higher solvent exposure (Fig. 2g). Also, binding of LSI-CoVA 014, 015, 016 may induce allosteric destabilization at RBM, which further needs to be probed (Fig. 2g).

Similar effects were observed across the other regions of Spike upon binding of LSI-CoVA-014, LSI-CoVA-015, and LSI-CoVA-016 (Supplementary Fig. 2b–d). Notably, peptides spanning residues 516–533 showed increased deuterium exchange (Fig. 2h, i, left panels), confirming that antibody-binding stabilized RBD in an up-conformation. This is accompanied by the loss of inter- and intra-monomer contacts between RBD and NTD, with residues 166–182 which interact with RBD, and 289–305 that connect NTD to the central Spike core showing the most significant changes (Fig. 2h, i, left panels). In the presence of LSI-CoVA-014, LSI-CoVA-015, and LSI-CoVA-016, an overall increased deuterium exchange was observed for peptides spanning the NTD regions of the Spike trimer. Further, multiple regions of the S2 subunit including the fusion peptide (FP), heptad repeats (HR1 and HR2) and residues 902–916 also showed increased deuterium exchange in the presence of these three HuMAbs (Fig. 2h, i, centre panels and Supplementary Fig. 3a–c), except the S1/S2 cleavage site which was associated with a decreased deuterium exchange. These sites are essential for inter-monomer interactions (Fig. 2h, i, right panels). Taken together, the conformational changes observed at the NTD and the S2 subunit suggest antibody-binding at RBD induces allosteric changes across the Spike trimer, resulting in its global destabilization that may lead to dissociation of adjacent monomers.

These observations were further supported by the HDX changes at the paratope sites of LSI-CoVA-014, LSI-CoVA-015, and LSI-CoVA-016 (Supplementary Figs. 3d, e and 4). Peptides spanning CDRL1-3 and CDRH1-3 showed greater differences in deuterium exchange in the RBD-bound complex, as compared to Spike-bound antibody complexes (Supplementary Table 2). The epitope sites are not hidden in the isolated RBD construct but are readily accessible to stably bind the paratope sites. On the other hand, in the Spike trimer, the RBDs must move from a down- to an up- position, and antibody binding is further hindered spatially by the NTD and the S2 subunit, leading to less stable antibody binding.

### NTD-binding LSI-CoVA-017 induces global stabilization of Spike trimer

We next investigated the effects of LSI-CoVA-017, which shows moderate neutralization. In the LSI-CoVA-017-bound state, protection against deuterium exchange was observed across the Spike trimer, with only a few peptides showing deprotection (Fig. 3a and Supplementary Fig. 5a). Upon closer examination, peptides spanning residues 92–110, 136–143 (N3 loop), and 243–265 (N5 loop) showed large-scale decreases in deuterium exchange in the LSI-CoVA-017-bound state (Fig. 3a). These peptides are positioned towards the outer edge of NTD, and are likely the epitope sites bound by LSI-CoVA-017. This

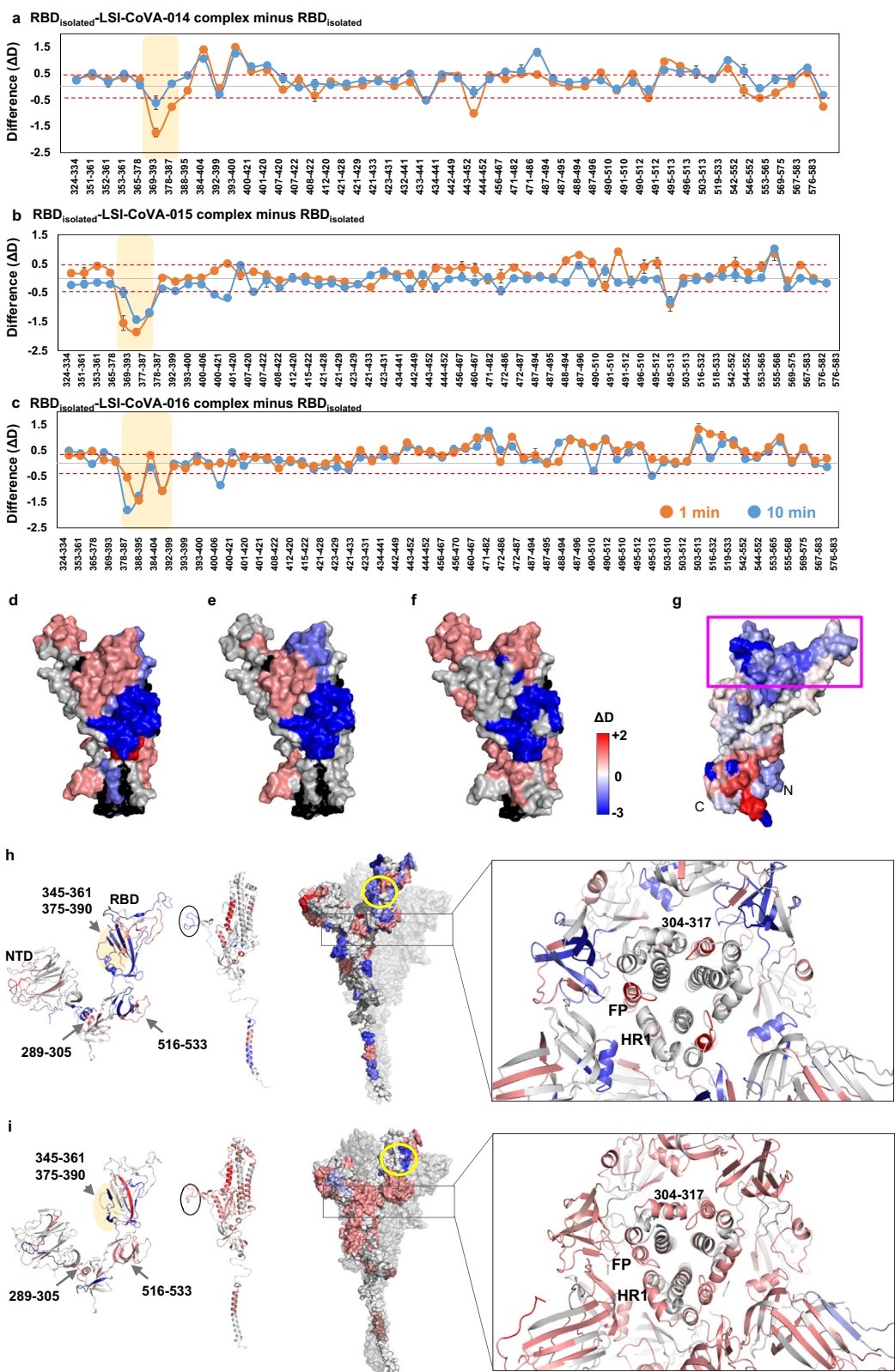

region also corresponds to the NTD antigenic supersite[20,39]. Reduction in deuterium exchange at short labeling times was observed across residues 36–48, 166–182, and 303–318, while increased deuterium exchange was observed for residues 60–83, 107–117, 213–228, and 266–276. These differences, mapped onto the structure of NTD (Fig. 3a, right panels), revealed that peptides encompassing the epitope site are clustered closely to form a structural epitope and

facilitate complexation of Spike with LSI-CoVA-017. These peptides also showed a reduction in deuterium exchange in our comparative HDX analysis of free Spike and its complex with 4A8, which has been previously characterized as an NTD-binding antibody[40] (Fig. 3b and Supplementary Fig. 5b). Thus, our binding assays and HDX data identify LSI-CoVA-017 as an NTD-binding antibody, with both LSI-CoVA-017 and 4A8 being moderate neutralizers.

**Fig. 2 | Mapping epitope sites for antibodies with weak neutralizing capacities.** Difference plots showing changes in deuterium exchange (ΔD) for **a** LSI-CoVA-014, **b** LSI-CoVA-015, and **c** LSI-CoVA-016 antibody complexes with isolated RBD compared to free RBD, at different labelling times as indicated. Pepsin-proteolyzed fragment peptides are represented by a dot and their residue numbers are indicated. Average values (*n* = 3 independent experiments) and the standard deviations are plotted using Microsoft Excel. A significant value of ± 0.5 D was considered as threshold and is indicated by red-dashed line. Epitope sites are highlighted in yellow. Differences in deuterium exchange values at 1 min labeling time for **d** LSI-CoVA-014, **e** LSI-CoVA-015, and **f** LSI-CoVA-016 antibody complexes are mapped on to the structure of RBD shown in surface representation, as per key. **g** The effects of ACE2 binding to RBD_Wuhan are mapped and shown for reference, with the RBM-site

highlighted in pink box. Comparative HDXMS analysis of Spike trimer (purified from insect cell culture) in the presence and absence of **h** LSI-CoVA-014 and **i** LSI-CoVA-015 and LSI-CoVA-016 highlighting S1 subunit (left), S2 subunit (centre), and Spike monomer (right, 1–1208 residues), with the other two monomers shown in grey. Peptides spanning key regions are highlighted by arrows. Epitopes on RBD (345–361, 375–390, 471–495) are indicated in yellow. Inset highlights a close-up view of Spike trimer (cartoon representation) along the transverse axis. Differences are mapped onto all the three monomers of Spike. RBD-binding antibodies (LSI-CoVA-014, LSI-CoVA-015, and LSI-COVA-016) induce destabilizing effects at the inter-protomer contacts. Peptides spanning residues 304–317, fusion peptide (FP), and heptad repeat 1 (HR1) constituting a part of intermonomer interaction interface are indicated. Source data is provided as Source data file.

Large-magnitude decreases in deuterium exchange were observed across all peptides (including residues 320-350, 516–533) spanning the RBD of Spike bound to LSI-CoVA-017 (Supplementary Fig. 5a and Supplementary Table 2). This indicates significantly reduced conformational dynamics across RBD, suggesting restricted domain motions in the LSI-CoVA-017-bound state. HDXMS analysis of LSI-CoVA-017 and 4A8 with isolated RBD, revealed no significant changes in deuteration levels of RBD (Supplementary Fig. 5a). Hence, it is clear that the antibodies binding at NTD induce distinct conformational changes across RBD and the S2 subunit compared to RBD-binding antibodies. Decreased deuterium exchange was observed for peptides spanning the S2 subunit of the Spike-LSI-CoVA-017 complex (Fig. 3c, d and Supplementary Fig. 5b). Upon LSI-CoVA-017 binding, notable changes in conformational dynamics were observed at the S1/S2 cleavage site, FP, central helix, and HR (Fig. 3c, inset). While both LSI-CoVA-017 and 4A8 binding resulted in similar effects on the Spike trimer, the changes induced by 4A8 HuMAb were less prominent. Overall, these HDXMS results reveal that LSI-CoVA-017 binding at NTD induces global stabilization of the Spike trimer.

HDXMS analysis of the LSI-CoVA-017 antibody showed significant changes across both heavy and light chains in the presence of Spike (Supplementary Fig. 5c and Supplementary Table 2). Peptides overlapping CDRH2 (residues 48–70), CDRH3 (96–103), and CDRL2 (48–71) showed protection against deuterium exchange, while CDRL3 (101–129) showed increased deuterium exchange. Interestingly, similar changes were observed for 4A8 complexed to Spike. No significant changes were observed for the light chain of 4A8 with or without Spike, consistent with available high-resolution structures[33].

The commonalities in effects of 4A8 and LSI-CoVA-017 upon Spike suggest similar modes of neutralization, as reflected in their neutralization capacities. However, our biophysical data showed LSI-CoVA-017 binds Spike trimer with an affinity much greater than 4A8 (Fig. 1b). To rationalize this, we determined the stoichiometry of the Spike-LSI-CoVA-017 complex by size-exclusion chromatography (Supplementary Fig. 6 and Supplementary Table 3). Three chromatographic peaks were detected and analyzed by denaturing polyacrylamide electrophoresis. Densitometry analysis of different amounts of peak B suggested a binding stoichiometry of three LSI-CoVA-017 antibodies per Spike trimer. With a 1:3 Spike:IgG stoichiometry, two models are plausible where: (i) Fab arms from three LSI-CoVA-017 antibodies bind to three monomers of a single Spike trimer; or (ii) two Fab arms of the same LSI-CoVA-017 bind monomers of two different Spike trimers. This is similar to the model predicted for the Spike:4A8 complex[33]. We further probed this computationally, as discussed below.

### Strong neutralizing HuMAbs bind at RBM and stabilize the Spike trimer conformation

Multiple studies have reported high-resolution structures of HuMAbs bound to RBM, including CoVA2-04, 5A6, and CoVA2-39, showing direct competition with ACE2 binding[38]. However, given that these HuMAbs display varying neutralization potencies in inhibiting viral

entry while binding to overlapping epitopes, a mechanistic explanation for their contrasting behavior remains elusive, particularly for CoVA2-04, a moderate neutralizer, as opposed to 5A6 and CoVA2-39 that are strong neutralizers. We therefore monitored the binding of CoVA2-04, 5A6, and CoVA2-39 to the Spike trimer and observed a distinct impact on its conformational dynamics (Fig. 4 and Supplementary Fig. 7). A large-magnitude decrease in deuterium exchange was observed across RBD, particularly the peptide clusters spanning RBM (485-502) of Spike complexes with 5A6, CoVA2-04 and CoVA2-39 (Fig. 4d, e and Supplementary Fig. 7a). Interestingly, HDXMS analysis of CoVA2-04 and CoVA2-39 complexes with the isolated RBD construct showed lower deuterium exchange across RBM, and only minor changes at other regions (Fig. 4a). These results indicate that binding of HuMAbs at RBM induces localized changes that lead to a significant reduction in the structural dynamics of RBD, including the peptides spanning the base and linker regions that connect RBD to the Spike trimer. Notably, the Spike variants contain mutations at different sites including E484K, N501Y- or K417N/E484K/N501 that are localized at RBM, and are reported to reduce the neutralization efficacy of antibodies[1,26,41].

Binding of CoVA2-04, 5A6, and CoVA2-39 to Spike resulted in increased deuterium exchange across peptide clusters covering residues 31–42, 92–110, 177–191, 265–276 of NTD (Fig. 4d, e and Supplementary Fig. 7a). Some of these peptides span the interface interacting with the RBD and the C-terminal region of NTD. As binding of these HuMAbs leads to RBD domain movement, it induces NTD movement as well, disrupting their interaction. Significant protection against deuterium exchange was observed at the S1/S2 cleavage site (residues 672–695), regions flanking the FP (residues 770–782, 878–898), HR1 (residues 927–962), central helix (1003–1031), and 1103–1117 of the S2 subunit (Fig. 4d, e, right panels, Supplementary Fig. 7b). These sites are essential for the Spike trimer to transition from its pre-fusion state to post-fusion state. Decreased deuterium exchange across the S2 subunit suggests that binding of these strong neutralizing HuMAbs leads to global reduction in the conformational dynamics of the Spike trimer, which may prevent the transition to the fusogenic intermediate. Collectively, the HDXMS results provide detailed insights into the mechanism of action of CoVA2-04, 5A6, and CoVA2-39, wherein they compete with ACE2 binding and induce stabilization throughout the Spike trimer to restrict its mobility.

### Glycan-F_ab interaction stabilizes Spike-antibody complexes

Next, we performed molecular docking to model the Fab domains of LSI-CoVA-014, LSI-CoVA-015, LSI-CoVA-016, and LSI-CoVA-017 at their respective epitope sites of the Spike protein (RBD or NTD) using HDXMS footprints as restraints (Fig. 5 and Supplementary Table 4), followed by atomic-resolution MD simulations. Out of 100 Fab:RBD/NTD complexes generated, five top-scoring binding poses were selected for 200 ns simulations (Fig. 5a–d). Among the simulated models of Fab:RBD/NTD complexes, multiple models were observed to either displace from the epitope site (Model 3, RBD-LSI-CoVA-014) or completely detach from RBD (Model 4, RBD-LSI-CoVA-014 and

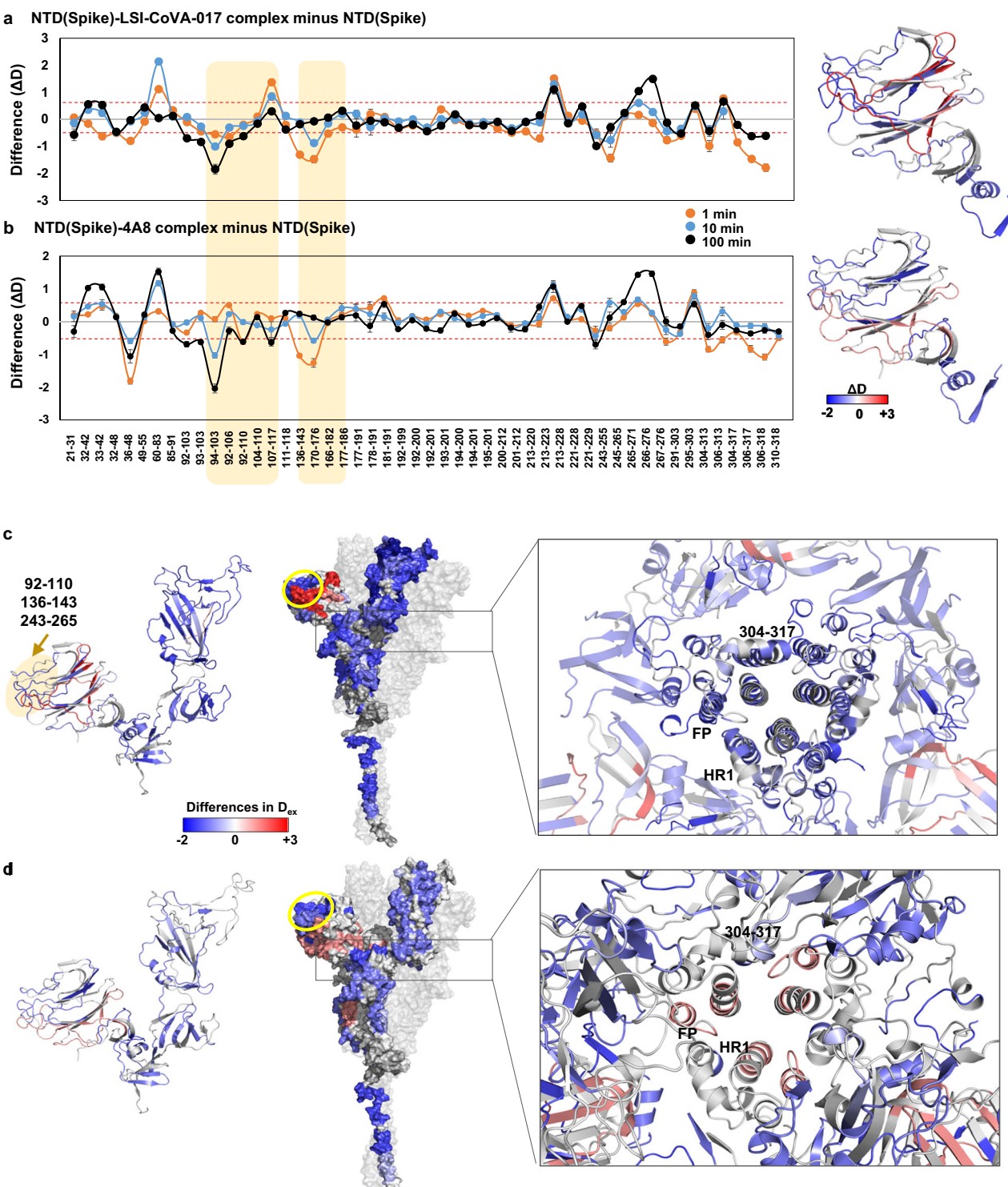

**Fig. 3 | Defining epitope sites for antibodies recognizing NTD.** Difference plots for NTD showing changes in deuterium exchange for **a** LSI-CoVA-017 and **b** 4A8 antibody complexes with Spike (purified from insect cell culture) compared to apo Spike, at different labelling times as indicated. Pepsin-proteolyzed fragment peptides spanning NTD of the Spike are represented by a dot and their residue numbers are indicated on the x axis. Average values ($n = 3$ independent experiments each for two biological replicates) and the standard deviations (error bars) are plotted using Microsoft Excel. A significant value of $\pm 0.5$ D was considered as threshold and is indicated by red-dashed line. Epitope sites are highlighted in yellow. Right panels: Differences at 1 min labeling time mapped on to NTD of Spike is shown in cartoon representation. Comparative HDXMS analysis of Spike trimer in the presence and absence of **c** LSI-CoVA-017 and **d** 4A8 mapped onto the S1 subunit (left) and trimeric Spike protein (right), as indicated. Peptides spanning key regions are highlighted by arrows with epitopes on NTD (92–110, 136–143, 243–265) indicated by yellow ellipse. Inset highlights a close-up view of Spike trimer along the transverse axis. Differences are mapped onto all the three monomers of Spike. NTD-binding LSI-CoVA-017 reduce overall conformational dynamics (shades of blue). Peptides spanning residues 304–317, fusion peptide (FP), and heptad repeat 1 (HR1) constituting a part of intermonomer interaction interface are indicated. Source data is provided as Source data file.

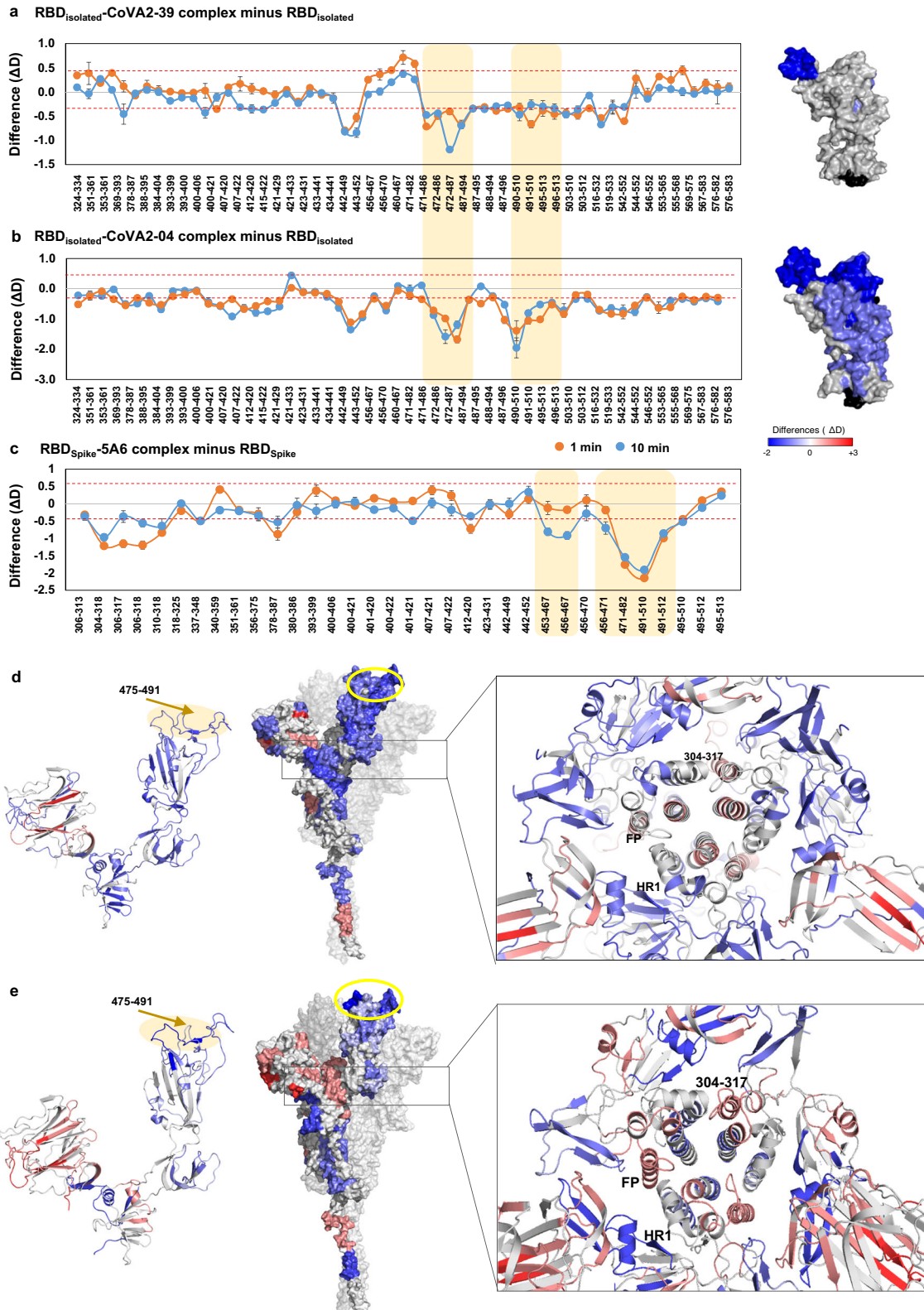

**Fig. 4 | Antibodies binding at RBM of Spike show moderate to strong neutralization.** Difference plots showing changes in deuterium exchange for **a** CoVA2-39 and **b** CoVA2-04 antibody complexes with isolated RBD, and **c** 5A6 antibody with RBD of Spike compared to apo RBD, at 1- and 10-min labelling times. Pepsin-proteolyzed fragment peptides spanning NTD of the Spike are represented by a dot and their residue numbers are indicated. Average values ($n = 3$ independent experiments each for two biological replicates) and their standard deviations (error bars) are plotted using Microsoft Excel. A significant value of $\pm 0.5$ D was considered as threshold and is indicated by red-dashed line. Epitope sites are highlighted in yellow. (right panels) Differences at 1 min labeling time mapped on to NTD of Spike is shown. Comparative HDXMS analysis of Spike trimer (purified from insect cell culture) in the presence and absence of **d** CoVA2-39 and **e** 5A6 mapped onto the S1 subunit (left), and Spike monomer (right), as indicated. Peptides spanning RBM epitope site are highlighted in yellow. Inset highlights a close-up view of Spike trimer along the transverse axis. Differences are mapped onto all the three monomers of Spike. Peptides spanning residues 304–317, fusion peptide (FP), and heptad repeat 1 (HR1) constituting a part of intermonomer interaction interface are indicated. Source data is provided as Source data file.

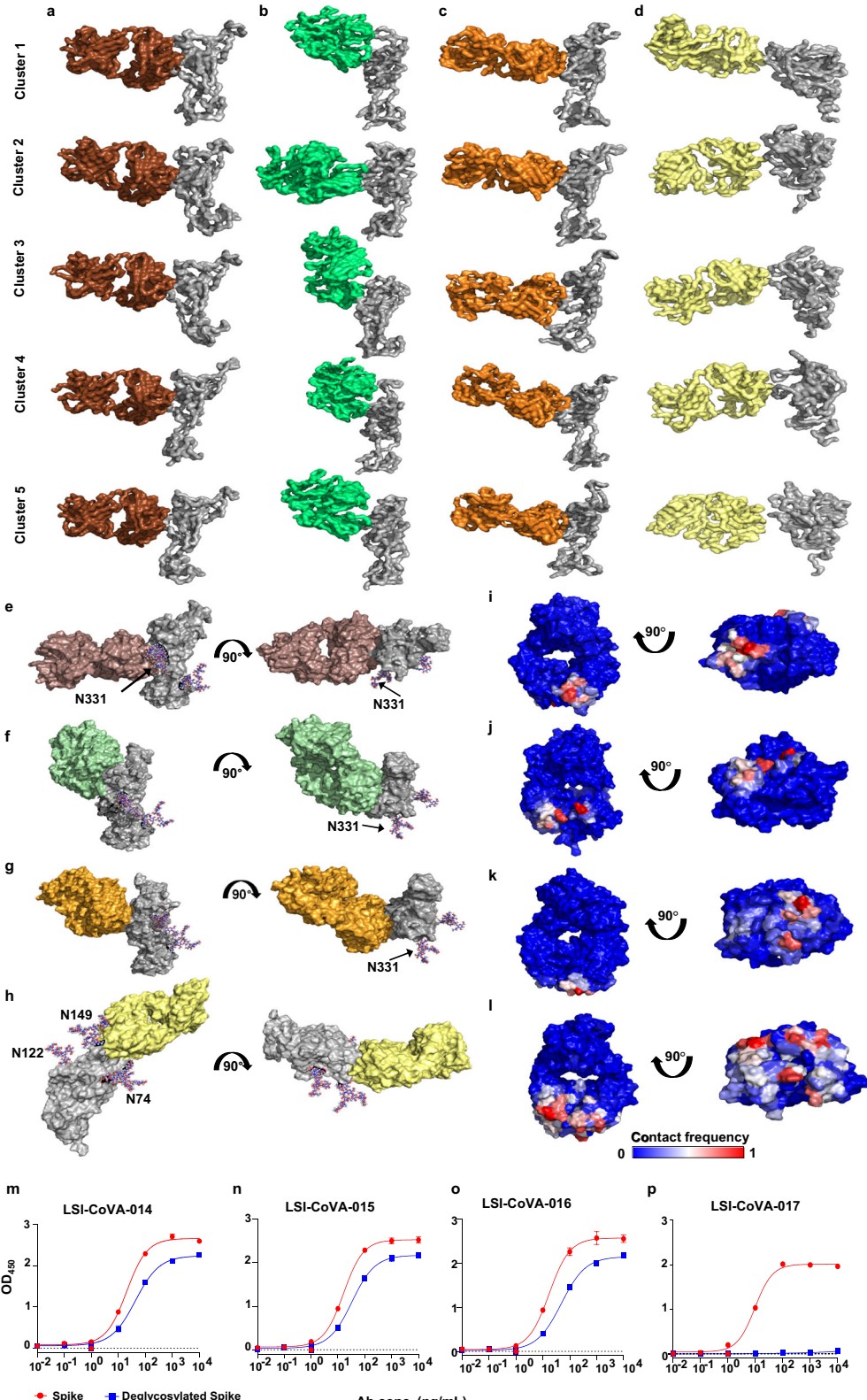

**Fig. 5 | Molecular dynamics simulations showing models of Fab bound to RBD/NTD and the essential role of glycans.** Surface representation of central structures of the five most populated clusters from **a** RBD-LSI-CoVA-014, **b** RBD-LSI-CoVA-015, **c** RBD-LSI-CoVA-016, and **d** NTD-LSI-CoVA-017 complexes. RBD and NTD are in gray, with LSI-CoVA-014 (brown), LSI-CoVA-015 (green), LSI-CoVA-016 (orange), and LSI-CoVA-017 (yellow). Representative structures from the most populated cluster from MD simulation trajectories are depicted in two orientations for RBD (grey) complexes with Fab of **e** LSI-CoVA-014 (brown), **f** LSI-CoVA-015 (green), **g** LSI-CoVA-016 (orange); and NTD (grey) in complex with **h** LSI-CoVA-017 (yellow). Glycans are shown in ball-and-stick representation. **i**–**l** Maximum contact frequencies between glycan moieties and Fab from simulations of top five RBD/NTD:Fab docking poses are mapped onto the structure of each Fab. Plots showing the binding of varying concentrations of **m** LSI-CoVA-014, **n** LSI-CoVA-015, **o** LSI-CoVA-016, and **p** LSI-CoVA-017 antibodies with Spike (red circles) and deglycosylated Spike (blue squares) as determined by ELISA. Data is represented as mean ± SEM ($n = 3$ independent experiments). Source data is provided as Source data file.

Model 3 of RBD-LSI-CoVA-016), and were not considered for further analysis (Supplemenary Fig. 8a). To select the best models from the stable complexes with each antibody, we next calculated the root mean square deviation (RMSD) of backbone atoms of the Fab domain (Supplementary Fig. 8b) and the model with the lowest RMSD was selected for additional replicate simulations to improve the conformational sampling (Supplementary Table 4 and Supplementary Fig. 9). A stable Fab binding orientation from the most populated cluster was identified via cluster analysis, of each Fab:RBD/NTD complex (Supplementary Fig. 9a) and the cluster analysis identified 34, 79, 52 and 65 clusters sampled for LSI-CoVA-014, LSI-CoVA-015, LSI-CoVA-016 and LSI-CoVA-017, respectively (Fig. 5e–h and Supplementary Fig. 9). Further, contact frequencies were calculated between the glycan moieties and Fab from RBD/NTD simulations of the top 5 docking poses and triplicate MD trajectories of selected poses, revealing that N-glycans interact with residues across Fab arms of LSI-CoVA-014, LSI-CoVA-015, LSI-CoVA-016 and LSI-CoVA-017 (Fig. 5i–l, Supplementary Fig. 8b, and Supplementary Fig. 9c). Interestingly, the glycan moieties at N343 of RBD and N74, N122 and N149 on NTD were observed to interact with the Fab (Fig. 5c). For the residues interacting with the N-glycans at N331 and N343, contact frequency maps showed the highest number and magnitude of contact frequencies made by LSI-CoVA-014 Fab, as compared to those by LSI-CoVA-015 and LSI-CoVA-016 (Supplementary Fig. 8b and Supplementary Fig. 9c). Similarly, simulation trajectories of the NTD-LSI-CoVA-017 complex showed prominent interactions between the N74, N122 and N149 N-glycans and the Fab (Supplementary Fig. 8b and Supplementary Fig. 9c). The contact frequencies measured for the NTD-LSI-CoVA-017 Fab complex indicated stable interactions with a larger surface compared to any of the RBD-Fab complexes, suggesting a potential role for glycans in forming the antibody epitope.

To verify this, we tested the binding of four novel HuMAbs ("LSI-CoVA") with a deglycosylated Spike trimer. Binding of LSI-CoVA-017 was completely abolished with deglycosylated Spike, in contrast to the minor changes observed for LSI-CoVA-014, LSI-CoVA-015 and LSI-CoVA-016 (Fig. 5m–p and Supplementary Fig. 10). Furthermore, significant reduction in binding kinetics of these four HuMAbs with deglycosylated Spike was observed, as compared to the glycosylated Spike trimer (Supplementary Fig. 10b). To further validate the significance of glycosylation to RBD-binding HuMAbs, we tested 5A6 as a control, which showed a partial reduction. Collectively, these results demonstrate that the LSI-CoVA-017 epitope encompasses glycan moieties on the Spike protein surface. For other antibodies (LSI-CoVA-014/LSI-CoVA-015/LSI-CoVA-016/5A6) the primary binding sites were non-glycosylated epitopes, as identified by HDXMS, with only secondary interactions contributed by glycans. These results provide a view contrary to the prevailing notion that glycans only act as a shield for Spike protein to hide epitope sites from host immune recognition[42,43] and suggest that non-specific interactions of glycans with the antibodies can play a substantial role in stabilizing Fab arm binding at the epitope site.

**Capture ELISA to assess combinatorial therapy**

A competitive ELISA was performed to evaluate the extent of epitope site overlap among antibodies (Fig. 6a–d) and also to characterize the cooperative binding to Spike monomers in the trimeric Spike. This allowed us to distinguish the mechanisms of binding and neutralization of RBD-specific antibodies that share the same or highly overlapping epitopes, yet have different affinities and neutralization activities. Competitive binding ELISA results indicated similar $OD_{450}$ values between LSI-CoVA-015 and LSI-CoVA-016 as detection or capture antibodies, suggesting a significant overlap in their binding orientation, which is in-line with our HDXMS-guided docking and MD simulations (Fig. 6e–j). On the other hand, LSI-CoVA-014 did not prevent binding of LSI-CoVA-015 or LSI-CoVA-016. Our simulation cluster

analysis of trajectories showed that LSI-CoVA-014 bound to Spike in a different orientation than LSI-CoVA-015 or LSI-CoVA-016, and thus could, in principle, pair with either LSI-CoVA-015 or LSI-CoVA-016 (Fig. 6e–h). Competitive assays between RBD- and NTD- recognizing antibodies showed that the binding sites for these two antibody classes do not overlap with each other, as observed for LSI-CoVA-014 and LSI-CoVA-017 with LSI-CoVA-015/LSI-CoVA-016 antibodies (Fig. 6a–c).

To infer stoichiometry and plausible mechanisms of neutralization, we then modelled the binding of full-length IgGs to Spike using a representative structure of Fab:RBD/NTD from the cluster analysis described above (Fig. 6e–h). Modelled full-length antibodies showed that RBD-binding antibodies specifically bind to RBD in the 'up'-position. Models of LSI-CoVA-015 and LSI-CoVA-016 complexes with the Spike trimer indicate that IgG binding to a single RBD of a Spike monomer sterically hinders the binding of a second IgG to the same Spike trimer. In the case of LSI-CoVA-014 and LSI-CoVA-017, the predicted orientation allows the respective full-length antibody to bind all three RBDs or NTDs of the same Spike protein trimer. These results are consistent with our competitive ELISA and neutralization assays. Additionally, the second Fab arm of LSI-CoVA-017 and LSI-CoVA-014 can bind to a second Spike protein trimer, cross-linking two Spike trimers (Fig. 6i, j and Supplementary Fig. 6). Taken together, the Spike-IgG complex models suggest that the novel antibodies characterized in this study indirectly interfere with ACE2 binding by either cross-linking Spike trimers on the viral surface (LSI-CoVA-014 and LSI-CoVA-017), or by blocking RBD-ACE2 interaction on a single Spike trimer (LSI-CoVA-015 and LSI-CoVA-016).

The four novel antibodies isolated here from convalescent patients showed suboptimal levels of neutralisation efficacy compared to RBM binding antibodies. However, considering the mutually exclusive epitope sites complemented by high affinity binding to Spike protein, it would be of interest to investigate their use in antibody cocktails. We explored the possibility to induce destabilisation in individual monomers or stabilization to reduce the hinge dynamics between the region connecting S1 and S2 subunits in order to effectively neutralise the SARS-CoV-2. Synergistic effects of selected HuMAbs used in this study were thus evaluated. The selected HuMAbs were used in a pairwise cocktail to study the potential synergistic enhancement of neutralization efficacy, amongst which the paired Mab cocktail of LSI-CoVA-017 and CoVA2-04 displayed a significantly higher percentage neutralization in comparison to the treatment of either of the single HuMAbs (Fig. 6d). We did not observe any enhancement in the neutralization efficacies of the two potent HuMAbs (CoVA2-39, 5A6) with NTD-binding LSI-CoVA-017. This could possibly be due to the limitation of the method and warrants alternative method for quantification purposes. Surprisingly, a combination of NTD- (LSI-CoVA-017) with RBD- (LSI-CoVA-014) antibodies resulted in lower neutralization, than when added alone.

**Cryptic site binding antibodies explain immune escape by Spike variants**

Emergence of new variants as a result of mutations of the Spike protein, have led to many antibody-mediated therapies faltering. Therefore, we assessed the impact of defined variant-linked mutations on binding and neutralization of the novel antibodies characterized in this study with isolated RBD and Spike proteins of the two former variants – Delta (δ) and Omicron (o1 for BA.1 and o2 for BA.2 lineages). LSI-CoVA-014, LSI-CoVA-015, and LSI-CoVA-016 bind to Spike and isolated RBD of all strains tested, although the binding activities to Omicron variants were slightly lower (Fig. 7), akin to CR3022 (Fig. 7). Binding of these antibodies was preserved as the mutations among the variants are distant from the cryptic epitopes site. Interestingly, 4A8 bound to Spike of all strains tested, although the binding activity was drastically reduced among all variants compared to Wuhan-Hu-1 Spike. On the other hand, the other NTD-binding antibody LSI-CoVA-017 bound only to Spike_Wuhan. This lack of binding of LSI-CoVA-017 and 4A8 to Spike_δ

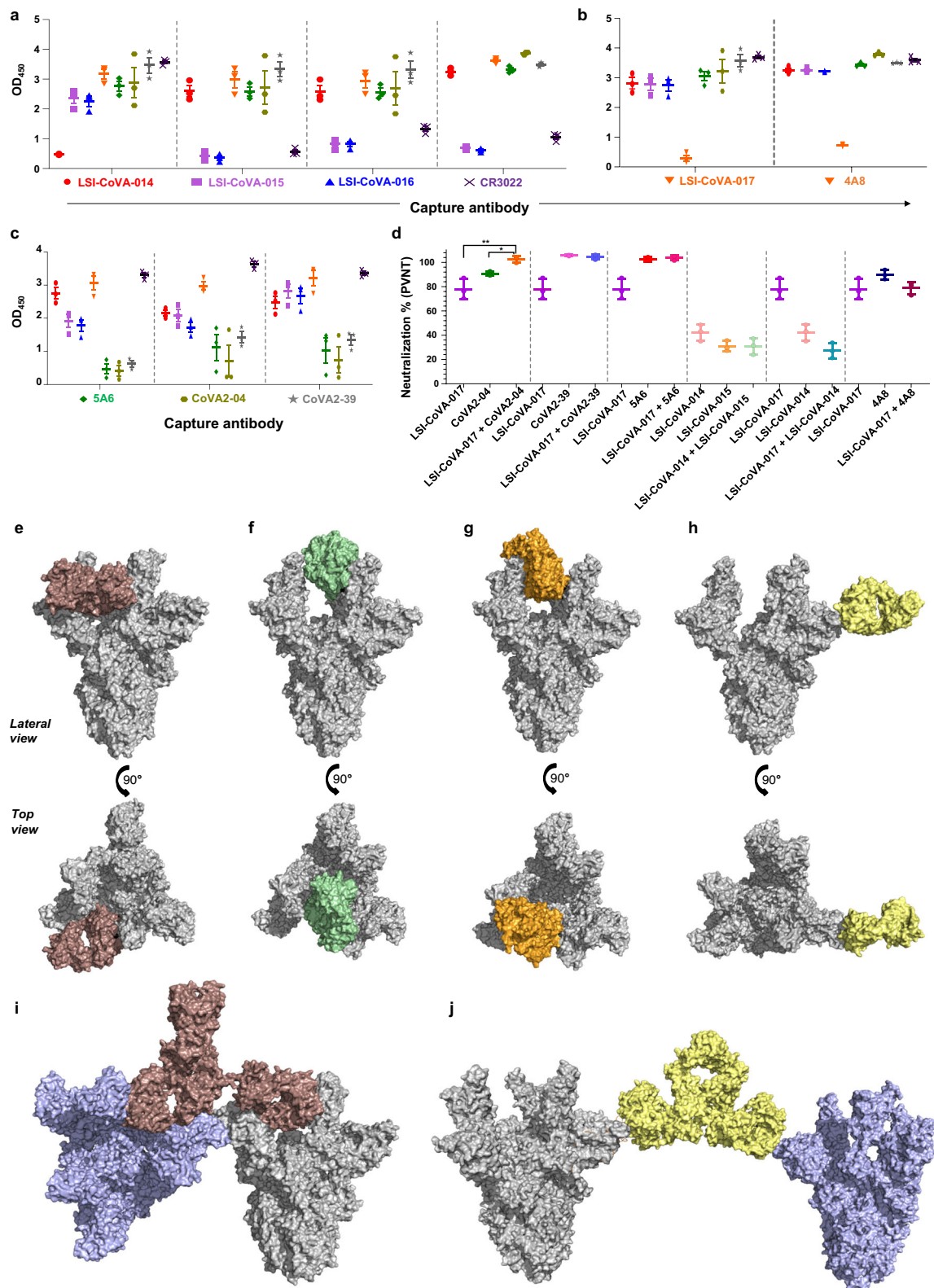

or Spike$_o$ is due to the deletions and mutations spanning the NTD antigenic supersite. Most importantly, 5A6, CoVA2-04, and CoVA2-39 which strongly bind and neutralize Spike$_{Wuhan}$, bound only to Spike$_\delta$ and RBD$_\delta$, but not the Omicron variants (Fig. 7a, open symbol plots). We also observed that upon deglycosylation, the binding activity of LSI-CoVA-017 and 4A8 was lost for all Spike variants, while minimal reduction was detected for anti-RBD antibodies against the Spike variants (Supplementary Fig. 10).

Many studies have reported higher binding affinities of Spike variants with the ACE2 receptor[44,45]. We therefore set out to probe if any of the nine HuMAbs competed with ACE2 binding to Spike variants. We performed ACE2-binding inhibition assays and observed a lack of any inhibitory activity by the cryptic site binders or the NTD-binding antibodies (Fig. 8a). Interestingly, LSI-CoVA-015 and LSI-CoVA-016 seemed to enhance ACE2 interaction with RBD$_{o1}$, as indicated by the negative inhibition (Fig. 8a, green arrows). These antibodies bind at

**Fig. 6 | Mechanism(s) of neutralization of Spike trimer Mabs.** Plots showing capture ELISA for pairs of selected antibodies. 0.1 µg Spike$_{Wuhan}$ (hexapro (purified from mammalian cell culture) was captured by **a** LSI-CoVA-014, LSI-CoVA-015, LSI-CoVA-016 and CR3022, **b** LSI-CoVA-017 and 4A8, and **c** 5A6, CoVA2-04 and CoVA2-39, detected using peroxidase-labelled monoclonal antibodies. A low OD$_{450}$ value is indicative of impaired binding of the peroxidase-labelled detection antibody, as listed. Data ($n = 3$ independent experiments) and represented as mean ± SEM. **d** Neutralization of pseudo SARS-CoV-2 virus using antibody cocktail (described in methods). Pair-mAb cocktail in the ratio of 1:9 to a final total concentration of 10 µg/ml or the single huMAb at a concentration of 10 µg/ml were incubated with pseudovirus lentiviral construct expressing the SARS-CoV-2 Spike protein. The chemiluminescence readout from the luciferase-tagged reporter in the lentiviral construct, is plotted and represented as percentage neutralization. The plots and

one-way ANOVA statistical analysis were done using GraphPad prism 9.0. *P* values determined were 0.0026 (LSI-CoVA-017 vs LSI-CoVA-017 + CoVA2-04) and 0.0339 (CoVA2-04 vs LSI-CoVA-017 + CoVA2-04) using 6 degrees of freedom, as indicated in the figure. Different antibodies are indicated by alternative colors. Data is reported as mean ± SEM ($n = 3$ independent experiments). Lateral (upper panel) and top (lower panel) views of surface representations showing Fab arms of **e** LSI-CoVA-014 (brown), **f** LSI-CoVA-015 (green), **g** LSI-CoVA-016 (orange), and **h** LSI-CoVA-017 (yellow) aligned onto a Spike trimer (grey, PDB 7A98). Representative structures from most populated cluster of Fab:RBD and Fab:NTD cluster analysis were used respectively. Predicted models showing two Spike trimers (grey and blue) bound to both Fab arms of **i** LSI-CoVA-014 (brown) and **j** LSI-CoVA-017 (yellow). Source data is provided as Source data file.

the cryptic site, and maintain RBD in an up-position, making the RBM site accessible, which may lead to increased ACE2 binding. This is similar to the effects of antibodies (e.g., S309) recognizing epitopes outside the RBM locus, and show some efficacy against the Omicron variant[46,47]. Also, 5A6, CoVA2-04, and CoVA2-39 inhibited interactions between ACE2 and Spike Wuhan/Delta strains, but this was not the case for in Spike$_{o1}$ or RBD$_{o1}$ (Fig. 8a, bottom panels). For ACE2 inhibition assays, neutralizers that bind RBD often exhibit >40% inhibition. Therefore, the binding and ACE2-inhibition results suggest that only the cryptic-site binding antibodies retain binding to Spike$_\delta$ and Spike$_{o2}$, and hence only their interactions were further explored.

## ACE2 binding enhances long-range conformational dynamics of Spike variants

We examined the effects of ACE2 binding on the conformational dynamics of the isolated RBD and trimeric Spike variants by HDXMS, and compared this with ACE2-binding footprints previously reported[5,44,48]. Binding of ACE2 elicited large-scale protection against deuterium uptake across all regions of isolated RBD$_\delta$, RBD$_{o1}$ and RBD$_{o2}$ (Fig. 8b). This altered conformational dynamics of RBD variants upon ACE2 binding is reflective of their higher binding affinities[44,48]. Despite this, mutations of key residues of RBM disrupted specific contacts between the RBD and ACE2, as reflected by available cryo-electron microscopy structures[45], and structural dynamics studies[49]. Upon closer examination of the HDX results, the ACE2 binding footprints were smaller for variants of RBD, as compared to RBD$_{Wuhan}$. Peptides spanning the mutation sites of the loop region (475-495) showed a lower degree of deuterium exchange, while residues 445-455 and 493-510 showed greater protection (Fig. 8b and Supplementary Fig. 11 a–c). We further determined the effects of ACE2 binding using trimeric Spike$_\delta$ and Spike$_{o2}$ (Fig. 8c, d and Supplementary Fig. 11d, e). Binding of ACE2 elicited conformational changes across RBD of Spike$_\delta$ (Fig. 8c, left, 8d) akin to those of isolated RBD$_\delta$ (Fig. 8b, left, 8d), as well as RBD of Spike$_{Wuhan}$. Surprisingly, we observed marked differences between the ACE2-bound states of RBD$_{o2}$ and Spike$_{o2}$ (Fig. 8c, right, 8e). While domain-wide decreased deuterium exchange was observed for the RBD$_{o2}$-ACE2 complex (Fig. 8b, right and Supplementary Fig. 11c), for RBD of the Spike$_{o2}$-ACE2 complex decreased deuterium exchange was observed only at residues 390–417 and 450–467 (Fig. 8e), and at 1 min labeling time for residues 488–507, as observed in high-resolution structures. Peptides spanning residues 373–384, 429–446, and 468–483 showed significantly increased deuterium exchange at all labeling time points, and residues 488–507 showed increased deuterium exchange at longer labeling times. Peptides showing deprotection overlapped the RBD-specific mutation sites observed for the Omicron variant, while increased protection against HDX was observed for Wuhan-Hu-1and Delta variants (Fig. 8c and Supplementary Fig. 11d, e). These results describe the molecular mechanism of ACE2-binding by the Omicron variant, whereby the specific amino acid residues promote receptor-binding by maintaining the essential conformational dynamics, yet evade immune responses. Furthermore, these HDX

findings also explain how ACE2 binding enhances the conformational sampling of variants, as observed by their high flexibility and fuzzy densities in cryo-EM maps[44,45] as well as the comparative structural dynamics of Spike variants containing the D614G mutation.

## LSI-CoVA-015 and LSI-CoVA-016 show cross-reactivity with Delta and Omicron variants

Using HDXMS, we also mapped and characterized the interactions of LSI-CoVA-014, LSI-CoVA-015, LSI-CoVA-016 antibodies with Delta and Omicron variants. Firstly, we characterized the interactions of the cryptic site binding antibodies LSI-CoVA-014, LSI-CoVA-015, and LSI-CoVA-016 with isolated RBD constructs of Delta (Supplementary Fig. 12), Omicron BA.1 (Supplementary Fig. 13), and Omicron BA.2 (Supplementary Fig. 14) variants. Consistent with our expectations, the most notable differences in deuterium exchange were observed for peptides covering the cryptic site, with LSI-CoVA-014, showing lower deuterium exchange values, compared to the LSI-CoVA-015 and LSI-CoVA-016 antibodies. These varying deuterium exchange values are reflective of the differences in conformations of RBD induced by variant-specific mutations. Further, we observed enhanced conformational dynamics for peptides spanning the ACE2-binding sites of the Delta and Omicron variants.

Next, we monitored the effects of these three antibodies binding to Spike$_\delta$ and Spike$_{o2}$ variants, which showed similar deuterium exchange profiles across the S1 (Fig. 9) and the S2 (Supplementary Fig. 15) subunits. Peptides flanking the mutated sites of NTD showed no significant change in deuteration as compared to Spike$_{Wuhan}$, which exhibited lower deuterium exchange upon binding to these three antibodies (Fig. 2). Importantly, the linker regions of NTD and RBD showed large-scale protection against deuterium exchange, due to reduced conformational flexibility, indicating that their domain motions were severely restricted. Specifically, peptides spanning residues 365–390 of RBD showed protection from deuterium exchange in the antibody-bound states of Spike$_\delta$ (Fig. 9a, b) and Spike$_{o2}$ (Fig. 9c, d), indicating that these three HuMabs bind at the same epitopes, akin to Spike$_{Wuhan}$. Upon closer examination, the magnitude of HDX changes across Spike$_{Wuhan}$, Spike$_\delta$ and Spike$_{o2}$ were different, owing to the differences in their binding affinities (Fig. 9a–d and Supplementary Fig. 15). Overall LSI-CoVA-015 and LSI-CoVA-016 (Fig. 9e, f, right panels) showed similar deuterium exchange values for peptides spanning the S1 subunit of Omicron Spike and were different compared to changes observed upon binding of LSI-CoVA-014 to Omicron Spike (Fig. 9e, f, left panels). Furthermore, HDX kinetics observed across the epitope sites of individual RBD variants (RBD$_\delta$, RBD$_{o1}$, and RBD$_{o2}$) in the presence of LSI-CoVA-015, LSI-CoVA-016, and LSI-CoVA-014, were stronger than their corresponding Spike trimers. This indicates that variant-specific mutations on Spike induce subtle changes in the conformational dynamics, which alter the binding strengths of antibodies even though the epitope sites are conserved. This is further supported by the varying HDX effects observed across peptides spanning RBM in antibody-bound Spike$_\delta$ and Spike$_{o2}$ (Fig. 9).

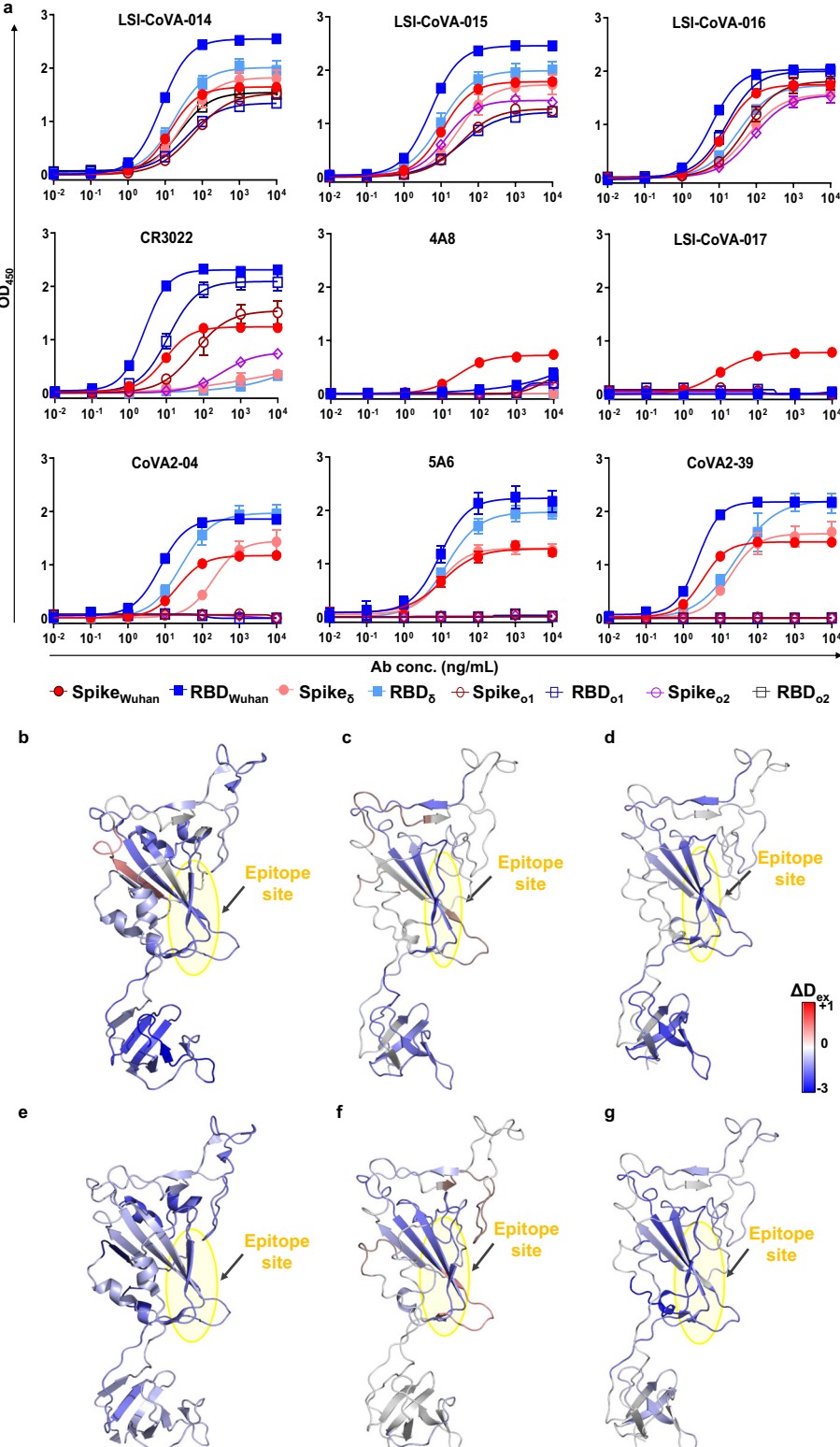

**Fig. 7 | Comparison of the binding activity of different human antibodies to Delta and Omicron variants. a** Antibody binding activity to Delta and Omicron variants of Spike compared with Wuhan-Hu-1 strain (HexaPro, purified from mammalian cell culture). Antibodies at concentrations from 10 pg/mL–10 μg/mL were tested for binding to SARS-CoV-2 Spike and MBP-RBD by ELISA, shown with their EC50 values indicated. Data was collected ($n = 3$ independent experiments) and represented as mean ± error bars (SEM). Heat map of differences in deuterium exchange for **b**–**d** LSI-CoVA-014 and **e**–**g** LSI-CoVA-016 antibody complexes with isolated RBD variants (**b**, **e**) RBD$_\delta$ (PDB: 7W98), **c**, **f** RBD$_{o1}$ (PDB: 7WPA), and **d**, **g** RBD$_{o2}$ (PDB: 7WPA) as compared to apo states. Cryptic epitope sites are highlighted in yellow. Source data is provided as Source data file.

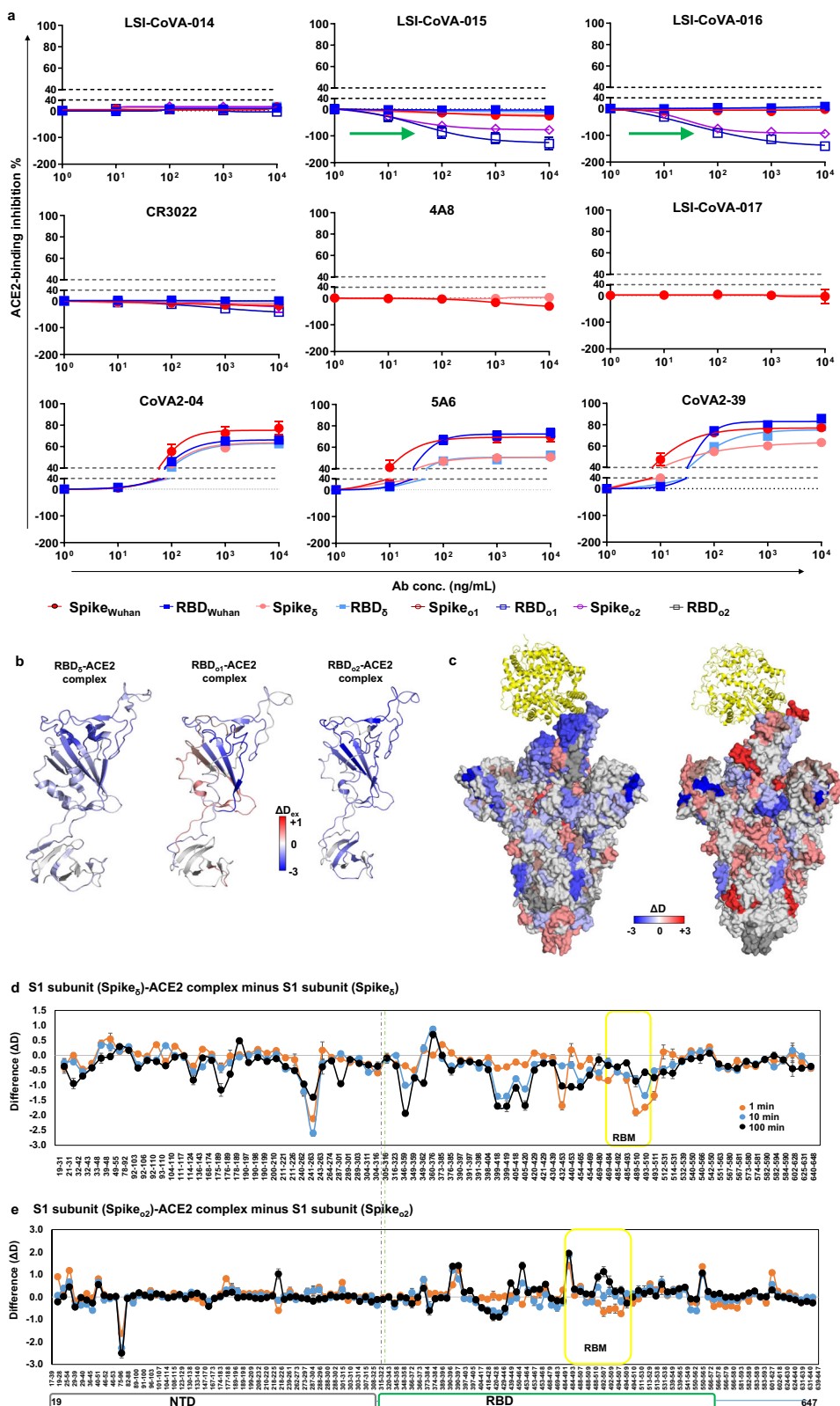

This effect was more prominent for RBD$_\delta$, RBD$_{o1}$ and RBD$_{o2}$, where binding of LSI-CoVA-014, LSI-CoVA-015, and LSI-CoVA-016 antibodies led to significant protection against deuterium exchange at the ACE2-binding sites (Fig. 9d). Our results show destabilization of the RBD of Spike$_\delta$ and Spike$_{o2}$upon antibody binding and explain why the up-position of RBD is favored in the variants[50].

Within the S2 subunit of Spike$_\delta$, the LSI-CoVA-014/015/016 deuterium exchange profiles (Fig. 9e, Supplementary Fig. 15b) were similar to that of Spike$_{Wuhan}$, but were significantly different from Spike$_{o2}$ (Fig. 9f and Supplementary Fig. 15a). Peptides spanning the S1/S2 cleavage site showed minor differences in deuterium exchange and were lower than those observed for Spike$_{Wuhan}$. The reduced

**Fig. 8 | Monitoring ACE2-binding to Delta and Omicron variants. a** ACE2-binding inhibition assays for the nine antibodies at varying concentrations for Spike (purified from mammalian cell culture) and RBD constructs of Wuhan-Hu-1(HexaPro), Delta (δ) and Omicron (o1, o2) variants. Negative values indicate enhanced ACE2 binding (green arrows), while positive values indicate inhibition of ACE2 binding by the antibody (Source data 8). Data was collected ($n = 3$ independent experiments) and represented as mean ± error bars (SEM). **b** Differences in HDX in the presence and absence of ACE2 for isolated RBD constructs of (left) Delta (PDB: 7W98), (centre) Omicron BA.1 (PDB: 7WPA), and (right) Omicron BA.2 (PDB: 7WPA) variants are mapped on to high-resolution structure of RBD, shown in cartoon. Shades of blue correspond to decreased deuterium exchange upon binding to ACE2. **c** Heat map of differences in deuterium exchange of (left) Spike Delta (PDB: 7W98) and (right) Spike Omicron BA.2 (PDB: 7WPA) in the presence and absence of ACE2 (yellow, cartoon) at 1 min labeling time is shown. **d, e** Plots comparing differences in deuterium exchange (ΔD) in the presence and absence of ACE2 for various peptides across the S1 subunit of Spike variants **d** Delta and **e** Omicron BA.2. Various labelling timepoints are indicated with peptide numbers are indicated in accompanying Source data. Data was collected ($n = 3$ independent experiments) and represented as mean ± error bars (standard deviation). Peptides covering NTD and RBD are grouped as per domain organization shown. ACE2-binding sites on RBM are highlighted in yellow. Source data is provided as Source data file.

deuterium exchange indicates that the S1/S2 cleavage site is more concealed in Spike_δ and Spike_o2, and is further occluded by antibody binding. These results also help to explain the reduced propensity for cleavage of variants and the inaccessibility to host proteases observed for the Omicron variant[29,51]. The most notable changes in conformation upon antibody binding were observed across peptide clusters spanning FP1, FP2 and HR1 (Supplementary Fig. 15). Large-magnitude increases in deuterium exchange were observed across these sites for Spike_δ and Spike_o2, as compared to the effects observed for Spike_Wuhan. Higher deuterium uptake correlates with increased dynamics and/or solvent accessibility across regions which span peptide clusters including residues 539–565 (subdomain 1), 757–779, and 936–974 (Fig. 9e, f right panels,). Increased deuterium exchange at these sites reflects greater solvent accessibility accompanied by the loss of intermonomer contacts. This translates to a long-range effect induced effect by LSI-CoVA-014, LSI-CoVA-015, and LSI-CoVA-016 across the Spike_δ and Spike_o2 trimers. Overall, these antibodies likely cause destabilization of the Spike trimers into antibody-bound monomers. This destabilization of the trimer, likely into individual monomers also reflects the altered interaction of ACE2 with Spike_o2 by LSI-CoVA-015/016, accompanied by weak neutralization efficacy.

### 5A6, CoVA2-39 and CoVA2-04 bind Spike_δ but not Spike_o2

Our binding assays indicated that 5A6, CoVA2-39, and CoVA2-04 bound to the Delta variant, but not the Omicron variant (Fig. 7a). We then compared the effects of binding of 5A6 and CoVA2-04 to Spike_δ and RBD_δ, using HDXMS. Upon binding of these HuMAbs, significant protection against deuterium exchange was observed across the RBDs, both in isolated constructs (Supplementary Fig. 16a) and in Spike_δ (Supplementary Fig. 16b, c). While the HDX changes observed across the RBM-sites of Spike_Wuhan were about ~4 Da, the average changes observed for the Delta variant were lower in magnitude (~2.5 Da) with a relatively smaller antibody-binding footprint. As the Delta variant has two key mutations at the RBM-site[26], the antibody footprint at the epitope site is reduced, affecting the overall conformational dynamics of the RBDs of Spike_δ. Although 5A6, CoVA2-39 and CoVA2-04 directly compete with ACE2 binding, various studies have reported that both CoVA2-39 and CoVA2-04 are unable to neutralize Delta or Omicron variants[52,53].

We further determined the effects of 5A6 (Supplementary Fig. 16b) and CoVA2-04 on the S2 subunit of Spike_δ. No significant change was observed for peptides covering the S1/S2 cleavage site. Decreased deuterium exchange was observed for peptides spanning FP2, HR1, connector domain (CD) and HR2. Importantly, peptides spanning the central helix (residues 990–1010), showed increased deuterium exchange in the presence of these two antibodies, suggesting higher localized conformational dynamics. This is in contrast to the effects observed for Spike_Wuhan (Fig. 4). While the sites essential for trimerization of Spike showed increased conformational rigidity induced by 5A6 and CoVA2-04 HuMAbs, increased conformational mobility was observed across the central helix. Collectively, the binding assays and HDX results indicate that the HuMAbs recognizing the RBM antigenic supersite cannot bind Omicron at all, and bind Spike_δ

variant with reduced affinity, by allosterically reducing the conformational dynamics of the S1 and the S2 subunit, thereby mediating overall stabilization.

## Discussion

Global efforts to tackle the COVID-19 pandemic produced effective mRNA vaccines, viral vector vaccines plus other innovations in diagnostic and preventive strategies aimed at reducing disease severity and mortality. However, the observed antigenic evolution of this virus linked to the emergence of escape variants translates into a potential for future outbreaks. Therefore, it is important to characterize the mechanism(s) of antibody-mediated neutralization to enable future therapeutic/prophylactic designs, as well as understand the molecular effects on the SARS-CoV-2 Spike protein. In this study, we have described antibodies principally derived from the memory B cell repertoire of convalescent COVID-19 patients and shown that these can be classified into three groups—weakly- (LSI-CoVA-014, LSI-CoVA-015, LSI-CoVA-016, CR3022), moderately- (LSI-CoVA-017, 4A8, CoVA2-04), and strongly- (CoVA2-39 and 5A6) neutralizing antibodies.

Importantly, we have highlighted the effects of binding of each of these antibodies to wild-type Spike (Wuhan-Hu-1) as well as Delta and Omicron variants, and provide mechanistic insights that distinguish neutralizing and non-neutralizing antibodies. Antibodies binding to the cryptic site on RBD (LSI-CoVA-014, LSI-CoVA-015, and LSI-CoVA-016) alter the local dynamics at the host ACE2 receptor-binding motif (Fig. 10a) and induce increased dynamics at distal sites across the Spike protein. Antibody-binding to RBD lead to significant changes in the conformational dynamics across the Spike trimer, in particular, binding at the cryptic site destabilizes the Spike trimer or induces formation of a previously described open trimeric conformation. However, this does not result in antibodies with augmented or stronger neutralizing activity. The highest neutralization capacity was observed for 5A6[34] and CoVA2-39 antibodies which bind at the ACE2 binding motif of RBD, and modulate Spike dynamics. While RBD has been the primary target for anti-SARS-CoV-2 antibodies, we characterized an NTD-binding HuMAb LSI-CoVA-017 and compared it to a well-characterized HuMAb 4A8[33,34]. Although these do not block virus attachment to ACE2, they still showed moderate neutralization, and their mechanism of action may involve cross-linking of two neighboring Spike proteins leading to antibody-mediated oligomerization (Fig. 10b–d). Further, stabilization of the Spike trimer and complexation was strengthened by interactions between the glycan groups on Spike with paratope sites of the mAbs. Additionally, blocking of NTD by antibodies also impairs binding to host receptors and favours neutralization[24,54]. Importantly, we used full-length IgGs (instead of isolated Fab domains) to capture their effects on the Spike protein. These are near native conditions and therefore provide physiologically relevant insights into the effects of avidity and restrained flexibility of the Fab domains (resulting from disulphide bonds in the IgG hinge regions). This also takes into account whether the Fab arms of IgG bind to the RBD of the same or different Spike trimer, which may induce different conformational dynamics. Further, HDXMS guided Fab:RBD/NTD docking and simulations revealed possible modes of IgG (LSI-CoVA-014, LSI-

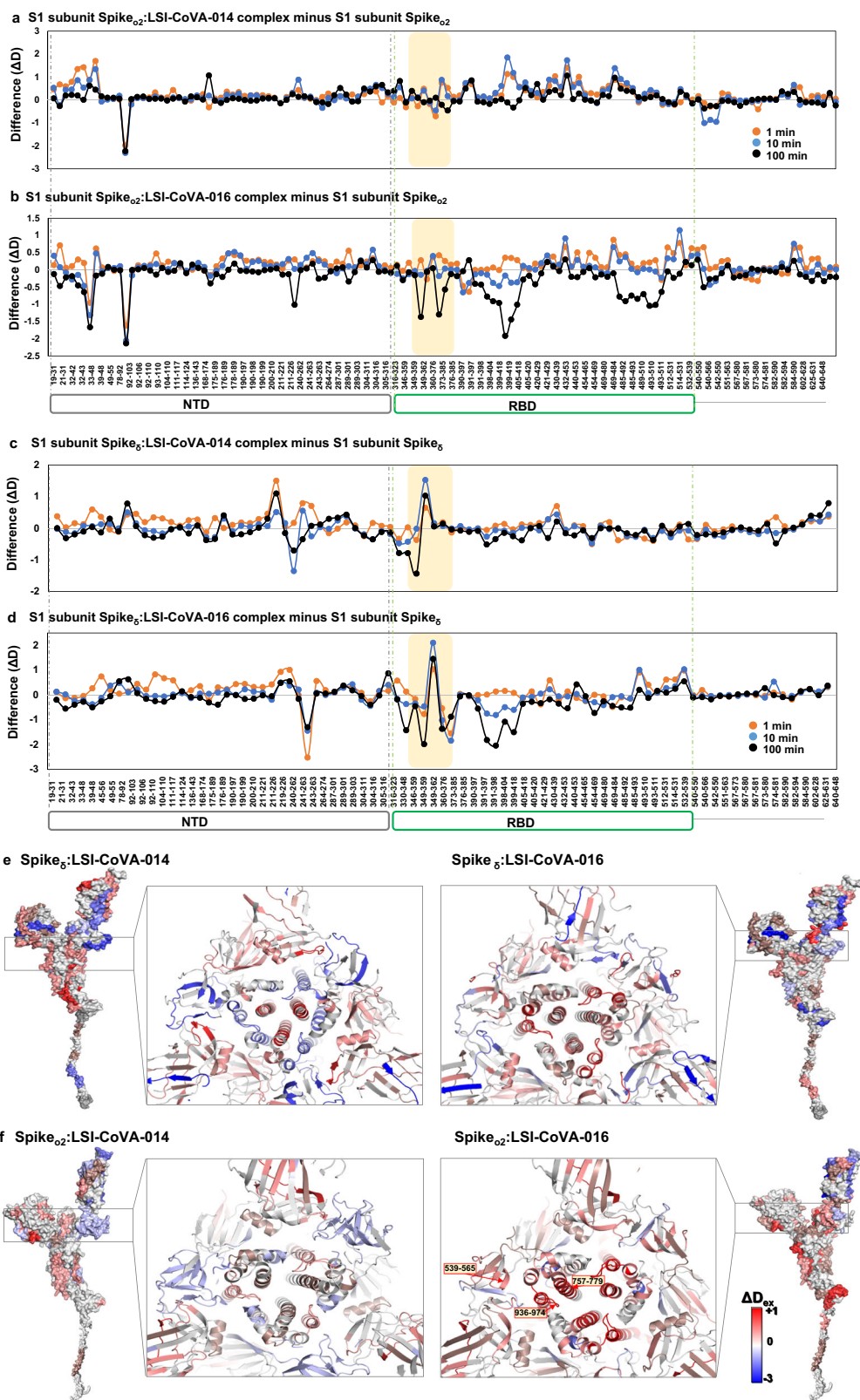

**Fig. 9 | Weak neutralizers bind to Spike variants at cryptic epitope site.** Plots comparing differences in deuterium exchange across the S1 subunit of a, **b** Spike$_{o2}$ and **c**, **d** Spike$_\delta$ complexed to **a**, **c** LSI-CoVA-014 and **b**, **d** LSI-CoVA-016 versus the apo states are shown at various labelling times. Residue numbers for the peptides are labeled on *x* axis of the bottom plots of **a**, **b** Spike$_{o2}$ and **c**, **d** Spike$_\delta$, with domain organization shown. Average (*n* = 3 independent experiments) values with standard deviations (error bars) were used to generate the plots (Microsoft Excel). Yellow zones highlight the peptides spanning the cryptic epitope site, as observed for

Spike$_{Wuhan}$. LSI-CoVA-016 showed larger decreases in deuterium exchange to the two variants. Differences in HDX of Spike variants—**e** Delta and **f** Omicron BA.2 in the presence and absence of (left) LSI-CoVA-014 and (right) LSI-CoVA-016 antibody are mapped onto a monomer of delta Spike (PDB: 7W98) and Omicron Spike (PDB: 7WPA), shown in surface representation. The insets show a transverse view of the conformational changes across the trimer. Source data is provided as Source data file.

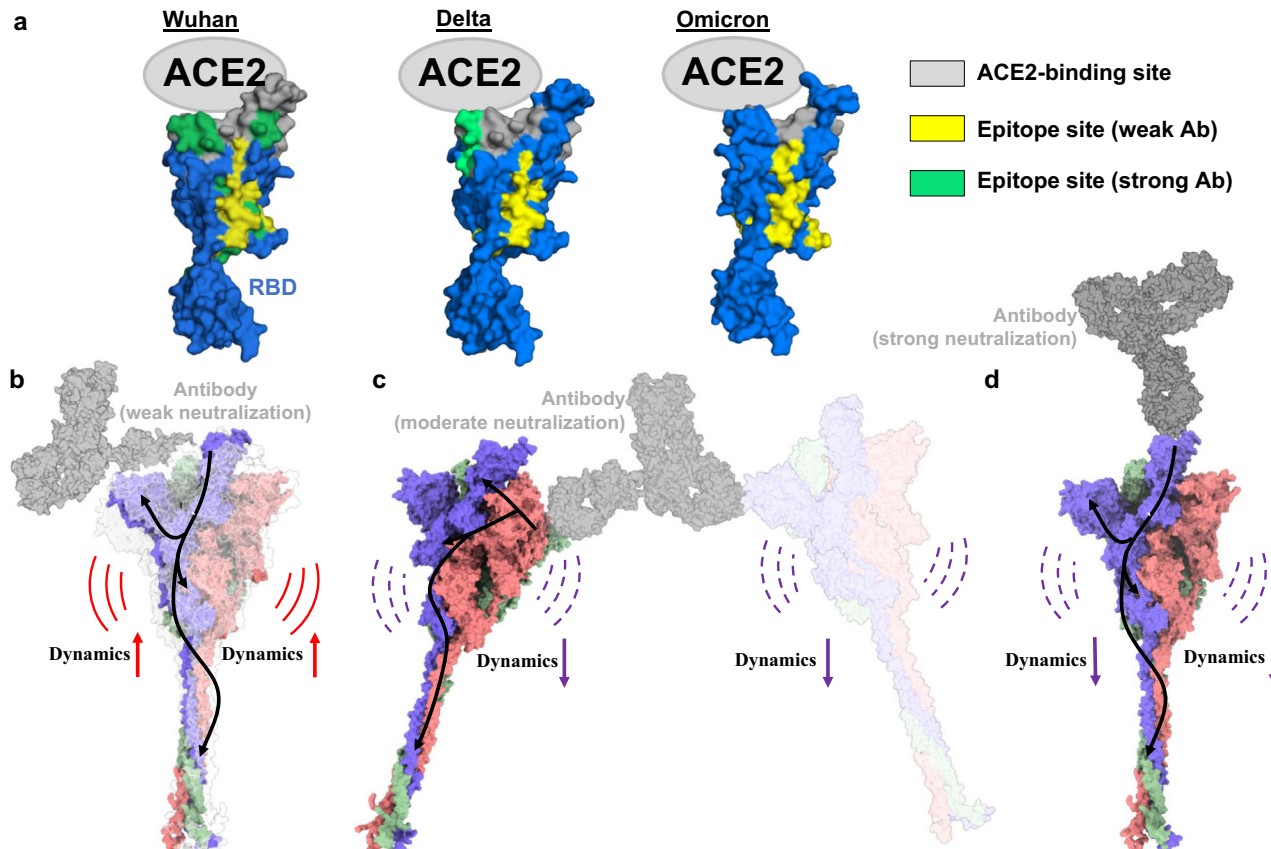

**Fig. 10 | Proposed mechanism(s) of antibody-mediated neutralization of Spike and its variants. a** Illustration showing the relative positions and binding footprints of ACE2 (gray), weak neutralizing antibodies binding to a cryptic site on RBD (yellow), and RBM-binding antibodies (green) for RBDs from Wuhan-Hu-1(left), Delta (centre) and Omicron (right) variants. Mutations across the variants lead to smaller footprints of ACE2 and impede binding of RBM-recognizing antibodies. However, antibodies that bind to the cryptic sites on RBD remain unaffected. **b** The different modes of neutralization of Spike trimer by the antibodies (IgG, gray) are illustrated. **b** Weak neutralizers bind to RBD in up position, which leads to allosteric relay (curved black arrows) across the S1 and S2 subunits and result in increased conformational dynamics (red arrows). These antibodies mediate neutralization via destabilization of the Spike trimer. **c** Moderate neutralizers or the NTD-recognizing antibodies bind across two Spike trimers and lead to decreased dynamics. **d** Strong neutralizers bind at the RBM supersite and directly compete with ACE2.

CoVA-015, LSI-CoVA-016 and LSI-CoVA-017) binding at the experimentally derived epitope sites. Even among the weak neutralizers that share the cryptic epitope site, the binding orientation of the Fab domain is different as confirmed by our IgG competition assay. Also, our simulations which were supported by ELISA identified the dependencies upon glycans of IgG binding to Spike protein. A recent study has shown that mutations in variants modulate Spike dynamics to enable escape from immunity, as well as improving pre-priming for fusion and entry into host cells[29].

The two orthogonal biophysical and biochemical approaches allowed functional characterization of antibody-binding to the Spike protein by epitope mapping and the accompanying conformational changes. The conformational dynamics elicited by the three different classes of antibodies are distinct from each other. The 'weak' antibodies show only a mild effect on the dynamics of the Spike trimer, inducing long-range destabilization/increased dynamics. On the other hand, the strong neutralizing antibodies showed large-scale conformational rigidity of the Spike trimer, primarily localized to the S1 subunit. The moderate neutralizers showed intermediate effects on the conformational changes of the Spike trimer, with both local and distal effects. Correspondingly, the weak-, moderate-, and strong-neutralization efficacies are reflective of the global changes on the Spike trimer. The changes in conformational dynamics were also influenced in some cases by interaction between the antibodies and the glycan moieties of Spike protein, as shown by both MD simulations and deglycosylation assays.

Additionally, our biophysical and biochemical characterization of both isolated RBD and the Spike trimer from Delta and Omicron variants has shed light on the probable reason why a potent neutralizing antibody of the Wuhan-Hu-1 strain was rendered ineffective against the emerging variants. Our HDXMS analysis of the Delta and Omicron variants reveal differences in their inherent protein dynamics compared to Wuhan-Hu-1, thereby reflecting their adaptation to evade host immunity. Comparative analysis of the isolated epitope domain (RBD) versus the complete target (Spike) revealed that antibodies targeting the Wuhan-Hu-1 strain were ineffective against the variants as expected. While the previously characterized antibodies (4A8, CoVA2-39, 5A6 and CoVA2-04) were unable to neutralize due to the mutations in RBD, the novel 'cryptic' site binding antibodies (LSI-CoVA-014, LSI-CoVA-015, and LSI-CoVA-016) identified in this study showed comparable binding to Delta and Omicron variants with respect to Wuhan-Hu-1. However, two main differences were observed between the isolated RBD and the full-length Spike for Delta and Omicron variants upon antibody binding to the cryptic site: (i) the conformational changes induced across Spike trimer were more significant than those observed for their corresponding RBD isolated constructs, indicating different modes of IgG binding to a single epitope (RBD) versus multiple epitopes (Spike); (ii) antibody binding elicited decreased flexibility across the C-terminus of isolated RBD constructs (as observed by significant protection against deuterium exchange (Supplementary Figs. 12–14)), but was not observed for the full-length Spike trimers. These results improve the understanding of evolutionary selection of

variants over other variants, wherein numerous mutations across Omicron variant evade known antibodies targeting the ACE2-binding site (strong neutralizers), but remain susceptible to antibodies targeting other sites of RBD, such as the cryptic epitope.

Further binding of LSI-CoVA-016 elicited greater effects on the S2 subunit rather than the S1 subunit of Omicron variant when compared to Wuhan-Hu-1 strain. Although the 'cryptic' epitope sites remain unchanged between these two strains, their conformational dynamics was significantly different. These results highlight how variant-specific dynamics can alter the binding strengths of antibodies, even at the conserved epitope site as well as across other regions of the Spike protein. Our analysis of the interactions of pairs of antibodies with Spike serves as a model to understand the physiologically relevant polyclonal conditions and antibody-induced additive effects that stabilize or destabilize Spike, which can result in either enhanced or decreased neutralizing efficacy. Determining binding of antibodies revealed which classes of antibodies allow simultaneous binding of more than one HuMAb to the Spike trimer.

Recently, bi-specific antibodies have been used against many targets, including SARS-CoV-2 Spike protein[55,56]. While there is a potential to use these against Omicron variant and restrict immune escape, not many bi-specific antibodies have been developed so far. The mechanistic insights highlighted in this study may allow specific reprogramming of the antibodies against non-overlapping sites of SARS-CoV-2 and minimize mutational escape. Moreover, there is no evidence supporting antibody-dependent enhancement of immunity by SARS-CoV-2 either in animal models or from clinical samples, unlike that observed for flaviviruses[57,58]. It is imperative to understand the effects of antibody binding to Spike protein and assess any suboptimal binding of new antibodies against variants. The distinct neutralization mechanisms by the three antibody categories also have implications in generating effective antibodies targeting the S2 subunit or other regions with low propensity for mutations, and allow effective antibody cross-reactivity between different strains/variants of SARS-CoV-2[31,55,59–61].

Consequently, the findings of this study provide insights for the rational design of next-generation of specific HuMAb cocktails of different classes of antibody to leverage synergistic interactions and mitigate the risks associated with emerging SARS-CoV-2 mutations, as well as preparing us for new coronavirus zoonoses in the future. Though the current study focuses on antibody modality, the mechanistic insights also support the parallel efforts of small molecule therapeutics at druggable sites on SARS-CoV-2.

## Methods

### Ethics statement
Human peripheral blood samples were obtained after informed consent from SARS-CoV-2 recovered patients (NHG DSRB Ref: 2020/00120, Establishment of assays for drug screening and virus characterization of the newly emerged novel coronavirus (2019-nCoV) which is also known as the Wuhan-Hu-1 coronavirus study). Study protocols were approved by the National University of Singapore Institutional Review Board. Participants were recruited after providing written informed consent to take part in the study under a DSRB approved protocol. All procedures performed involving human participants complied with relevant ethical regulations.

### Sorting of memory B cells
Cryopreserved human peripheral blood mononuclear cells (PBMCs) derived from blood samples from COVID-19 convalescent patients that were thawed at 37 °C in Isocove's Modified Dulbecco's Medium (IMDM) + Glutamax (Gibco) supplemented with 10% ultra-low IgG FBS (Gibco) and 1% penicillin/streptomycin (Gibco), 50 μg/mL human transferrin (Sigma-Aldrich), and 5 μg/mL human insulin (Sigma-Aldrich), was used for culturing of human memory B cells. The medium is referred to as complete IMDM. Memory B cells were then isolated using the EasySep™ Human Memory B cells isolation kit as per manufacturer's protocol (STEMCELL™ Technologies).

### Isolation and cloning of SARS-CoV-2 Spike, RBD-specific human antibodies
The sorted human memory B cells were then resuspended in complete IMDM with the addition of an activation cytokine milieu, containing the following: 20 U/mL IL-2 (Cell Guidance Systems), 50 ng/mL IL-10 (Cell Guidance Systems), and 10 ng/mL IL-15 (Cell Guidance Systems) and 50 ng/mL monomeric soluble recombinant human CD40L (eBioscience). The memory B cells were cultured for 4 days under these conditions to allow for stimulation and expansion. After 4 days, the cells were pelleted and resuspended in new complete IMDM with the addition of a secretion cytokine milieu, containing the following: 20 U/mL IL-2, 50 ng/mL IL-10, 10 ng/mL IL-15, and 50 ng/mL IL-6 (Cell Guidance Systems). This was to promote differentiation to antibody-producing plasmablasts. Following another 3 days of culture, the cells were pelleted, supernatants used for screening of binding activity on SARS-CoV-2 Spike by ELISA, and pellets lysed using the QuickExtract RNA extraction kit (Lucigen). The supernatants were harvested and used for ELISA screening. All cytokines were purchased from Cell Guidance Systems, USA.

Cell lysates, corresponding to the positive hits from the ELISA screen, were then used for downstream PCR and NGS analysis. cDNA was generated using the Maxima H Minus cDNA Synthesis Master Mix (Thermo Fisher) as per the manufacturer's protocol. Obtained cDNA products were directly used for separate amplification of the heavy and light-chain variable regions. Illumina adaptor and barcode sequences were added by further PCR and the PCR products were purified with AMPure XP magnetic beads. Purified PCR products were pooled to obtain a 4 nM library and sequenced on the Illumina MiSeq with a 2 × 300 bp kit with 25% PhiX spike-in. Individual reads were separated and corresponding germline sequences were retrieved from IMGT, the international ImMunoGeneTics database. The individual heavy and light sequences obtained used for plasmid synthesis.

### Antibody and antigen expression and purification
Genes coding for variable regions of antibody heavy and light chain were synthesized and cloned into vector pTT5 expression vector (National Research Council Canada, NRCC) by Twist bioscience. HEK 293-6E (NRCC, Cat #11565) cells were cultured in freestyle f17 expression medium (Life Technologies). Antibody heavy and light chain constructs were transfected into cells at a concentration of 400 ng/ml in Branched-PEI (Polyethylenimine, Sigma-Aldrich, Cat #408727)/150 mM NaCl at 1:3 (DNA:branched PEI) ratio. 7 days post-transfection, culture supernatant containing the expressed antibodies was harvested via centrifugation, filtered through a 0.22 μm filter unit (Merck, SG) and loaded onto a MabSelect Sure column (Cytiva, SG) to purify the antibody. The purified antibody was subjected to buffer exchange using Vivaspin centrifugal concentrator (Sartious, GE) and concentrated in 1× PBS, pH 7.2.

Gene encoding for near full-length Spike (S) proteinSARS-CoV-2 (16-1208; Wuhan-Hu-1; GenBank: QHD43416.1) and containing RRAR (residues 682–685) mutated to DDDDK and KV(residues 986-987) mutated to PP was codon optimized for insect cell expression and cloned into pfastbac expression vector (Biobasic, SG). Gene was followed by a trimerization domain, HRV 3 C site, 8x-Histidine tag, and a streptavidin tag at the C-terminus. SARS-CoV-2-Spike construct was expressed in *Spodoptera frugiperda (Sf9)* (Thermo Fisher Scientific, Singapore Cat # 11496015) cells following instructions from bac-to-bac baculovirus expression system (Thermo Fisher, SG). Briefly, bacmid of SARS-CoV-2 Spike was generated, purified, and used for transfection using cellfectin II (Thermo Fisher Scientific, SG). Viral stocks obtained from transfection were amplified and used for protein expression.

Culture supernatant was harvested by centrifugation at day 4 post infection. Spike protein was affinity purified using a 5 mL HisTrap excel column (Cytiva, SG). Purified Spike was concentrated using a 100 kDa cut-off concentrator (Sartorius, GE) and loaded onto a HiLoad 16/60 superdex 200 pre-equilibrated with 20 mM Hepes, pH 7.5, 300 mM NaCl, 5% glycerol. Peak fractions corresponding to the trimer size of Spike protein were collected and concentrated. SARS-CoV-2-RBD was expressed and purified as described previously[5]. Plasmid containing gene encoding SARS-CoV-2 Spike HexaPro (1-1208) was obtained from AddGene. It contains six proline mutations (F817P, A892P, A899P, A942P, K986P, V987P) and the furin site (682–685) was replaced with 'GSAS'. Plasmid DNA was transfected into Expi293 cells (Thermo Fisher Scientific, SG, Cat #A14527) using expifectamine following manufacturer's protocol. Culture supernatant was harvested on day 7 post transfection. Spike hexapro was purified following the same protocol as described for Spike construct expressed by *sf*9 cells. Spike and RBD variant sequences were obtained from Global Initiative on Sharing All Influenza Data (GISAID) database. Genes encoding Spike variants (Delta and Omicron BA2) with Spike hexapro mutations incorporated were synthesized and cloned into the hexapro plasmid to replace the original hexapro gene (Biobasic, SG). Spike variants were purified using the same method as Spike hexapro. Spike trimer purified from *sf*9 cells was mainly used for HDX analysis, glycosylated vs deglycosylated Spike binding affinity to antibodies by QCM, while Spike hexapro trimer purified from mammalian cell culture was used for other binding analysis. Human ACE2 used for HDX analysis was purified as described previously[5].

RBD variants were made using KLD enzyme mix (NEB, SG) with RBD wild-type construct[5] as the template, except for RBD Omicron variant, for which the gene was synthesized (Biobasic, SG) and cloned into the same template to replace the wild-type RBD using Gibson assembly master mix (NEB, SG). Variant-specific mutations were introduced and verified by sequencing. The list of primers used for inducing mutations is tabulated in Supplementary Table 5. RBD variants were expressed and purified as described before[5]. The purity and integrity of the Spike and RBD variants and the antibodies were determined by denaturing polyacrylamide electrophoresis (Supplementary Fig. 17).

### Antibody binding activity analysis using enzyme-linked immunosorbent assay (ELISA)

Spike was coated at 100 ng/well and RBD at 200 ng/well onto 96-well flat-bottom maxi-sorp binding immunoplates (SPL Life Sciences, SG) and incubated overnight at 4 °C. The following steps were conducted at room temperature. Plates were washed three times in PBST (phosphate buffer with 0.05% Tween 20) and blocked with 350 μL of blocking buffer (4% skimmed milk in PBST) for 90 min. A tenfold serial dilution was performed for each primary antibody to get seven concentrations ranging from 10 μg/mL to 0.01 ng/mL. Plates wash step was repeated and the diluted primary antibodies were added at 100 μL/well. Reference wells with no primary antibody (blocking buffer only) were included. After 60 min incubation, plate wash step was repeated and the detection antibody, goat anti-human IgG-HRP (Thermo Fisher, SG) was added at 100 μL/well for an incubation of 60 min in dark. This was followed by the plate wash step and an incubation with 1-Step Ultra TMB-ELISA (Thermo Scientific, SG) at 100 μL/well for 3 min. Reaction was stopped with 100 μL 1 M $H_2SO_4$ per well. Optical density at 450 nm was measured using a microplate reader (Tecan Sunrise, SG). Each antibody was tested in three independent replicates and their average values were used to generate the plots.

### Biophysical characterization of binding of mAbs to Spike

Trimeric Spike or PNGase F treated Spike HexaPro (0.03 μM in 10 mM sodium acetate buffer, pH 4.5) was immobilized onto the surface of an LNB-Carboxyl quartz crystal chip (Attana, SE) using sulfo-NHS/EDC. Unbound carboxyl groups were quenched using ethanolamine. A

reference chip was similarly activated and quenched, but with no protein bound to its surface. Chips were stabilized in the Attana Quartz Crystal Microbalance (QCM) instrument and experiments performed at 22 °C using a flow rate of either 20 μL/min or 25 μL/min. Running buffer was 10 mM HEPES, 150 mM NaCl, 0.005% Tween 20, pH 7.4 (HBST). Injections of 35 μL, were made in triplicate ($n$ = three independent experiments) over both chips, and both association and dissociation phases of each interaction were recorded for 380 seconds using Attana software. Experiments were performed on a variety of ligand densities to control for avidity. Reference curves were subtracted using Attester v2.0.0.57 software and the resulting curves imported into TraceDrawer v1.9.1 (Ridgeview Instruments) for kinetics evaluation.

### SARS-CoV-2 pseudotyped lentivirus production

Pseudotyped viral particles expressing SARS-CoV-2 Spike proteins were produced using a third-generation lentivirus system. A reverse transfection methodology was used for this assay. At day 1, $36 \times 10^6$ HEK293T cells were transfected with 27 μg pMDLg/pRRE (Addgene, US), 13.5 μg pRSV-Rev (Addgene, US), 27 μg pTT5LnX-WHCoV-St19 (SARS-CoV-2 Spike) and 54 μg pHIV-Luc-ZsGreen (Addgene, USA) using Lipofectamine 3000 transfection reagent (Thermo Fisher, SG) and cultured in a 37 °C, 5% $CO_2$ incubator. On day 4, supernatant containing the pseudoviral particles was harvested and filtered through a 0.45 μm filter unit (Merck, SG). The filtered pseudovirus supernatant was concentrated using 40% PEG 6000 by centrifugation at $1600 \times g$ for 60 min at 4 °C. Lenti-X p24 rapid titre kit (Takara Bio, JP) was used to quantify the viral titres, as per manufacturer's protocol.

### Pseudovirus neutralization assay (PVNT)

On day 0, CHO cell lines with stable expression of ACE2 (a kind gift from Prof. Tan Yee-Joo, Department of Microbiology and Immunology, National University of Singapore), were seeded at a density of $5 \times 10^4$ cells in 100 μL of complete medium [DMEM/high glucose with sodium pyruvate (Thermo Fisher, SG), supplemented with 10% FBS (Thermo Fisher, SG), 10% MEM non-essential amino acids (Thermo Fisher, SG), 10% geneticin (Thermo Fisher, SG) and 10% penicillin/streptomycin (Thermo Fisher, SG)] in 96-well white flat-clear bottom plates (Corning, US). Cells were cultured in 37 °C with humidified atmosphere at 5% $CO_2$ for one day. The next day, the respective monoclonal antibodies (mAbs) were serially diluted ten times in sterile 1× PBS. The diluted samples were incubated with an equal volume of pseudovirus to achieve a total volume of 50 μL, at 37 °C for 1 h. The pseudovirus-antibody mixture was added to the CHO-ACE2 monolayer cells and left incubated for 1 h to allow pseudotyped viral infection. Subsequently, 150 μL of complete medium was added to each well for a further incubation of 48 h. The cells were washed twice with sterile PBS. 100 μL of ONE-glo™ EX luciferase assay reagent (Promega, SG) was added to each well and the luminescence values were read on the Tecan Spark 100 M. The percentage neutralization was calculated as follows:

$$\text{Neutralization\%} = \frac{\text{Readout(unknown)} - \text{Readout(infected control)}}{\text{Readout(uninfected control)} - \text{Readout(infected control)}} * 100\%$$

(1)

### Neutralization by antibody pairs

Pair-mAb cocktail in the ratio of 1:9 to a final total concentration of 10 μg/mL, or single HuMAb at a concentration of 10 μg/mL were incubated with pseudovirus lentiviral construct expressing the SARS-CoV-2 Spike protein. The antibody:pseudovirus mixtures were then added to CHO-ACE2 cells. The chemiluminescence readout from the luciferase-tagged reporter in the lentiviral construct, was then plotted and represented as percentage neutralization. The assay was

conducted in triplicate. The data is shown as mean ± SD GraphPad Prism was used for plots and one-way ANOVA statistical analysis.

## Determining stoichiometry of LSI-CoVA-017 binding to Spike

80 μg spike protein and 100 μg LSI-CoVA-017 (1:1.25 molar ratio) were mixed and incubated at room temperature for 30 min, prior to injection on a Superose 6 Increase 10/300 GL column (GE Healthcare, SG) in buffer (20 mM Tris, 200 mM NaCl, pH 8). The resulting chromatographic eluted peaks were analyzed by SDS-PAGE. Densitometry analysis was carried out using Image Lab (v6.1.0 build 7, Bio-Rad, SG). Briefly, 1–4 μg of Spike and LSI-CoVA-017 were loaded on SDS-PAGE as quantification standards. 40 μL and 25 μL of peak A, and 100 μL and 50 μL of peak B were loaded on SDS-PAGE for quantification. The absolute quantities of Spike, heavy and light chains of LSI-CoVA-017 were estimated based the band intensities relative to the quantification standards. For each lane, the quantities of LSI-CoVA-017 were then derived from the quantities of either heavy chain or light chain. Both methods estimated similar quantities of LSI-CoVA-017, and thus consistently suggest a binding stoichiometry of three LSI-CoVA-017 bound per Spike trimer. In addition, consistent results of the two different loading amounts per peak A or peak B provided further confidence in the reliability of the densitometry (refer to Supplementary Table 3 for detailed calculation).

## Hydrogen-deuterium exchange mass spectrometry (HDXMS)

Purified Spike trimer (10 μl of 8 μM) and isolated RBD (1.5 μl of 67 μM), solubilized in aqueous PBS (pH 7.4) were diluted in 20× deuterated PBS to attain 90% final $D_2O$ concentration[5]. The deuterium labelling was performed for 1, 10, 100 min for Spike protein, and 1 and 10 min for isolated RBD respectively. Similarly, HDX was performed for nine convalescent antibodies– LSI-CoVA-014, LSI-CoVA-015, LSI-CoVA-016, and LSI-CoVA-017 identified in this study; and CR3022, CoVA2-04, CoVA-39, 4A8, and 5A6 (previous studies) individually. All the deuteration reactions were performed at 37 °C. The same experimental setup was done for isolated RBD and Spike constructs of Delta and Omicron variants.

For HDXMS of antibody–protein complexes, saturating concentrations of antibody to the antigen were used. For each antigen-binding site on HuMAb was considered as one antigenic site, and mixed with Spike trimer in 1:3 (protein:antibody), and 1:1 isolated RBD:antibody stoichiometry. As the antibodies bind with a high affinity, the antigen-antibody complexes were incubated for 30 min at 37 °C to achieve >90% binding, before each deuterium exchange reaction. The resultant protein:antibody complexes (-100 pmol) were diluted in 20x deuterated PBS (90% final concentration of $D_2O$) and subjected to 1, 10, and 100 min hydrogen-deuterium exchange timescales. For Delta and Omicron variants, the stoichiometric ratios were similar to the Wuhan-Hu-1 strain. Deuterium exchange of Spike variants with host ACE2 receptor were carried out by mixing saturating amounts of ACE2 at 37 °C, as described earlier[5].

At the end of labelling times, each deuteration reaction was quenched by adding pre-chilled quench solution (1.5 M Guanidinium-HCl, 0.25 M TCEP-HCl) to lower the $pH_{read}$ to -2.5 and incubated at 4 °C on ice for 1 min. The quenched samples were injected onto nanoUPLC™ HDX sample manager (Waters, USA) for proteolysis by immobilized pepsin cartridge (Enzymate BEH pepsin column 2.1 × 30 mm (Waters, USA)) for 7 min, with a continuous flow of 0.1% formic acid in water (UPLC grade, Merck, GE) at 100 μl/min. The pepsin proteolyzed peptides were trapped on a 2.1 × 5 mm C18 trap (ACQUITY BEH C18 VanGuard Pre-column, 1.7 μm, Waters, USA) and then eluted by a gradient of 8-40% of 0.1% formic acid in acetonitrile using reverse phase column (ACQUITY UPLC BEH C18 Column, 1.0 × 100 mm, 1.7 μm) pumped at a flow rate of 40 μl/min by nanoACQUITY binary solvent manager (Waters, USA). Total trapping (3 min) and elution (7 min) was 10 min long. Peptides were ionised using electrospray

ionisation method and sprayed onto Synapt G2-Si mass spectrometer (Waters, UK) as described previously[5]. During the gradient elution, injector and pepsin-column were subjected to wash using wash solution (1.8 M Gn-HCl, 8% acetonitrile, 0.4% formic acid). Between triplicate deuterium-exchange measurements, a wash injection was performed to eliminate any carry-over.

Protein Lynx Global Server (PLGS) v3.0 was used to identify the detected peptides from mass spectra of undeuterated samples. A separate sequence database of Spike protein (Wuhan-Hu-1, Delta, Omicron BA.1, and Omicron BA.2 variants, UniProt accession P0DCT2), and human ACE2 protein (Q9BYF1) along with the respective purification tags were used to identify pepsin proteolyzed peptides, as described previously[5]. Identified peptides with mass (MH +) tolerance of <10 ppm, minimum intensity–2000, maximum peptide length–25, minimum product per amino acid–0.1 were filtered and then analysed for deuterium uptake using DynamX v3.0 (Waters, USA). The deuterium exchange for each peptide across all labelling time points for each protein state (apo, antibody-bound) were manually verified before final analysis. The deuterium uptake was estimated as the difference in the masses of the centroids of the deuterium-labeled peptide and the corresponding unlabeled peptide. Deuterium exchange experiments were performed in biological and technical replicates and average values are reported as HDX summary (HDX Summary Table in Supplementary Information), and are not corrected for deuterium back exchange. As differences in deuterium exchange data are used to analyze and interpret the conformational changes, instead of deuterium uptake, the HDX values are not corrected for back exchange. HDX data are shown as difference plots generated using DynamX v3.0 and Microsoft Excel 2016, as heat maps generated using PyMol (Schrodinger), and Deuteros 2.0[62]. The fractional and relative deuterium uptake values for various peptides of Spike and isolated RBD constructs of Wuhan-Hu-1, Delta, and Omicron in bound- and free- states are tabulated in Source data and supplementary information.

## Homology modelling of single Fab arm

Homology models of Fab arms of LSI-CoVA-014, LSI-CoVA-015, LSI-CoVA-016, and LSI-CoVA-017 were built using Modeller version 9.21[63]. Position-specific iterative-BLAST was used to identify the template structures with high sequence identity and structures available on the PDB were chosen. PDB 7K8R, 7DWZ, 6DF2, and 7JXE were used as template structures to model peptides spanning heavy chain and light chain of LSI-CoVA-014, LSI-CoVA-015, LSI-CoVA-016, and LSI-CoVA-017 respectively. Heavy (1–226) and light (1–216) chains were modelled separately along with intramolecular disulphide bonds. For each antibody, 100 models were generated, and the model with the lowest discreet optimized protein energy (DOPE) score was chosen to generate a Fab arm. Hetero dimers containing heavy and light chain for respective antibodies were complexed by aligning to the respective template structures in PyMol (Schrodinger Inc, USA). Corresponding full-length IgGs were modelled using Swiss Modeller server using 1IGT and 1MCO structures as templates (sequence identity >90%).

## Glycosylated RBD, NTD modelling and antigen–Fab complex generation

Atomic coordinates of RBD and NTD from full-length Spike protein (PDB 6XR8[64]) were extracted and further modelled. Glycan chains were constructed using CHARMM-GUI[65] Glycan Reader and Modeller[66] at the reported glycosylation sites i.e., N331 and N343 of RBD, and N17, N61, N74, N122, N149, N165, N234, and N282 of NTD[67,68]. The glycan composition reported in a previous study was used to model the glycans at the respective glycosylation sites[67]. The epitope sites on RBD and NTD identified using HDXMS for each antibody were chosen to perform a biased docking using ClusPro 2.0 webserver[69]. The peptides showing protection in the presence of antibody were provided as input under attractive amino acid residues at the protein-protein interaction

interface in ClusPro, using a specific module for antigen-antibody docking. Ten different poses or orientations of Fab bound to RBD/NTD were generated at the epitope site involving complementarity determining regions (CDR) of Fab arms. Each pose was inspected using VMD[70] and the top five poses with the highest docking score were chosen to perform atomistic MD simulations from each Fab:RBD/NTD complex.

## Simulation setup and protocol

Atomistic MD simulations were performed to identify a structurally stable orientation of Fab bound to RBD and NTD. A total of 20 Fab:RBD/NTD complexes from four different Fab arms were modelled using CHARMM-GUI and simulated in GROMACS package version 2018.4[71]. Each system was parameterized using the CHARMM36m force field[72]. Each system was solvated with TIP3P water molecules and neutralizing 0.15 M NaCl. The heavy atoms in the Fab:RBD/NTD complex were restrained using a force constant of 1000 kJ mol$^{-1}$ nm$^{-1}$ to perform a 125 ps equilibration simulation. A 310 K temperature was maintained using the Nóse-Hoover thermostat with 1.0 ps time constant and 1 atm pressure was maintained using isotropic coupling to the Parrinello-Rahman barostat with time constant of 5.0 ps[73,74]. Electrostatic interactions were calculated using the smooth particle mesh Ewald's method with a real-space cut-off of 1.2 nm. The van der Waal's interactions were truncated at a distance of 1.2 nm with a force switch smoothing function imposed from 1.0 to 1.2 nm. The LINCS algorithm[75] was used to constrain all the covalent bonds formed with hydrogens with an integration time step of 2 fs. A 200 ns long production run was performed for all the 20 models. The best model for each of the four Fab arms were then selected based on the lowest backbone RMSDs. Two further 200 ns independent repeat simulations with different starting velocities were performed for each of these models. The list of simulations performed are provided in Supplementary Table 4.

Most analysis was performed using the GROMACS package. Pairwise distances between atoms of glycans and Fab molecules were calculated and any pairwise distances less than 0.4 nm were recorded as an interaction. Furthermore, to identify the stable or dominant orientation of Fab bound to RBD and NTD, we performed a cluster analysis using the GROMOS method with an RMSD cut-off of 0.35 nm. The central structure from the most populated cluster was chosen to perform further analysis and modelling. The trajectories of model 3 and model 4 from RBD-LSI-CoVA-014 and model 4 and model 3 from RBD-LSI-CoVA-016 were not considered for cluster analysis.

## Capture ELISA

Monoclonal human IgG1 antibodies LSI-CoVA-014, LSI-CoVA-015, LSI-CoVA-016, and LSI-CoVA-017 were conjugated individually to peroxidase as per manufacturer's protocol (Abnova, TW). The final concentrations of conjugated antibodies are 1 mg/mL. Unconjugated LSI-COVA-014, LSI-CoVA-015, LSI-CoVA-016, and LSI-CoVA-017 were diluted in PBS to a final concentration of 10 μg/mL, and coated on 96-well maxisorp binding immunoplates at 100 μL/well for overnight incubation at 4 °C. Plate wash and blocking steps were conducted as mentioned above. SARS-CoV-2 Spike protein at 1 μg/mL diluted in blocking buffer was added to each well, 100 μL/well for 1 h incubation. Plate wash was repeated. Peroxidase-conjugated antibodies were added individually at a final concentration of 1 μg/mL (diluted in blocking buffer) for 1 h incubation protected from light. All following steps were performed as mentioned above.

## Binding activity of antibodies to PNGase F treated Spike

In total, 20 μg of SARS-CoV-2 Spike trimer was deglycosylated by incubating with 2.5 μL of PNGase F (NEB, SG) under native condition at 37 °C for 4 h. Spike trimer and deglycosylated Spike trimer were coated at 1 μg/ml, 100 μL/well on 96-well maxisorp binding immunoplates for 1 h at room temperature. Following steps of the ELISA are the same as described above. Monoclonal antibodies LSI-CoVA-014, LSI-CoVA-015,

LSI-CoVA-016 and LSI-CoVA-017 were tested at several concentrations ranging from 0.01 ng/mL to 10 μg/mL.

## ACE2-RBD-binding inhibition ELISA

The neutralizing capacity of antibodies can be directly correlated to their ability to inhibit the interactions between ACE2 and Spike-RBD (Variants). The percentage inhibition was evaluated using an ELISA-based surrogate viral neutralization test. The Wuhan-Hu-1 and Delta RBD antigens were coated at 1 μg/mL and the Omicrons BA.1 and BA.2 were coated at 4 μg/mL onto 96-well flat-bottom MaxiSorp immunoplates (SPL Life Sciences) for 30 min at room temperature. The plates were then washed thrice using an automated washer (Biotek EL 406) with washing buffer (1× PBS with 0.05% Tween-20) and blocked for 60 min with blocking buffer (3% bovine serum albumin in washing buffer). The respective mAbs were then serially diluted 10 times in blocking buffer. In addition, the negative controls (5x diluted heat-inactivated FBS in blocking buffer) and positive controls (100 μg/mL ACE2-FC in 5x FBS) were prepared for each RBD variant. Following the incubation period, the washing step was repeated. The mAbs, negative and positive controls were added onto the plates for 45 min at room temperature. Following the primary incubation, the washing procedure was repeated. The plates were incubated with the HRP-conjugated ACE2-Fc detector antibody at 200 ng/mL for the Wuhan-Hu-H1 and Delta variants and at 600 ng/mL for the Omicron variants, for 45 min at room temperature and protected from light. The wash step was repeated and TMB substrate (Thermo Scientific) was added. After 3 min of incubation, 1 M $H_2SO_4$ was added to stop the reaction and the optical density at 450 nm ($OD_{450}$) was then recorded using a microplate reader (Tecan 100 M). The formula below was used to calculate the percentage inhibition values.

$$Inhibition(\%) = \frac{Readout(negative\,control) - Readout(sample)}{Readout(negative\,control)} \times 100 \tag{2}$$

## Reporting summary

Further information on research design is available in the Nature Portfolio Reporting Summary linked to this article.

## Data availability

The authors declare that all data supporting the findings of this study are available within the paper and its supplementary information files. Data analyzed and used to generate the plots is included as Source data and Supplementary Information, provided with this paper. The materials or reagents used in the analysis is available upon request. Raw mass spectrometry data generated at SingMass facility, with the derived data available from the corresponding author upon request. The mass spectrometry HDXMS data generated in this study have been deposited to the ProteomeXchange Consortium via PRIDE partner repository using the dataset identifier: PXD043818. P0DCT2 (Spike), Q9BYF1 (human ACE2 protein) accession codes were used. PDB codes used in this study are 7K8R, 7DWZ, 6DF2, 7JXE, 1IGT, 1MCO, 6XR8, 7A98, 7W98, 7WPA. A HDX summary table is included as supporting table, as per community-acceptable standards.

## Code availability

Simulations input data and coordinates files have been deposited to the Zenodo public repository and can be accessed via the following link: https://doi.org/10.5281/zenodo.8354172.

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

## Acknowledgements

We would like to acknowledge A/Prof Ganesh S. Anand (Department of Chemistry, PennState University, USA) for his initial inputs. We would like to thank Ms. Eve Zi Xian Ngoh for her contribution in discovery and production of 5A6 antibody. CHO cell lines with stable expression of ACE2 was a kind gift from Prof. Tan Yee-Joo, Department of Microbiology and Immunology, National University of Singapore. This work used computational resources of the National Supercomputing Centre (NSCC), Singapore (https://www.nscc.sg), the A*STAR Computational Resource Centre (A*CRC), and the supercomputer Fugaku provided by RIKEN through the HPCI System Research Project (Project ID: hp220297) awarded to P.J.B. and F.S. This study is supported by COVID-19 (R-571-000-081-213) and SCOPE (R-711-000-058-598) grants awarded to P.A.M. by National Medical Research Council, Singapore; FY21 CG HTPO SEED ID BII C211418001 funded by A*STAR awarded to P.J.B., and AME YIRG (A2084c0159) grant funded by A*STAR awarded to F.S. R.V.P., F.S., and P.J.B. were supported by BII (A*STAR) core funds. We acknowledge SingMass, Singapore for mass spectrometry facility for mass spectrometry data acquisition.

## Author contributions

Conceptualization: N.K.T., R.V.P., P.J.B., and P.A.M. Methodology: N.K.T., R.V.P., Q.X., G.Y., B.S., F.S., L.J., W.Y.H., B.W., M.M.K., and K.P. Investigation: N.K.T., R.V.P., M.M.K., J.L., C.W., P.J.B., and P.A.M. Visualization: N.K.T., R.V.P., X.Q., G.Y., and L.J. Funding acquisition: P.A.M. and P.J.B. Project administration: M.M.K. and P.A.M. Supervision: N.K.T., J.L., C.W., P.J.B., and P.A.M. Writing—original draft: N.K.T., R.V.P., X.Q., L.J., and R.G. Writing—review & editing: N.K.T., R.V.P., X.Q., F.S., R.G., J.L., P.J.B., and P.A.M.

## Competing interests

The ACE2-RBD-binding inhibition ELISA has been commercialized and branded by GENY BIOLOGICS as the ImTracker MULTI COVID-19 viral variant neutralization test (https://genybiologics.com/imtracker/)[76,77]. The following authors are co-inventors: P.A.M., B.S., and G.Y. The remaining authors declare no competing interests.
