## [Peer Review File · Nature Communications]

Reviewers' Comments:

Reviewer #1:

Remarks to the Author:

Tulsian et al presents new data about how antibodies lead to binding- induced allosteric effects on the conformation of Spike and antibody. The main method is HDXMS to determine dynamics in binding. The authors tested nine mAbs, four of them were newly generated in this manuscript. First affinity+ kinetic and neutralization of these mAbs was tested and the mAbs were divided into weak, moderate and strong neutralizers. By HDXMS effect on RBD and or NTA-Spike of the binding – but only for six of the nine mAbs- was analysed and mapped on structures. Capture ELISA was performed to get combinational mAbs pairs. Some combinations were also tested in neutralization. Next binding to Wuhan, delta and Omicron of six of the nine mAbs was tested in ELISA, ACE2-binding inhibition assay and some of them in differenced in deuterium exchange. A model of how the different mAbs effect and neutralize was obtained. Glykan binding was also analysed but data can be only found in the supplement.

As I am not in expert in structural biology (modelling) nor on HDXMS in I cannot comment on the quality of these data or on the material and method section.

In total, the manuscript comprises a lot of impressive data and experiments, interesting for the scientific community- BUT for some passages restructuring/rewriting (incl. reordering of figures) is advised as the structure/story is not clear and the reader is lost - this might also be caused by the density of data, which in general is a positive aspect but it should be appropriately packaged. Other major limitations are that some results remain unexplained (examples see below).

Discussion in general is quite short and not really a discussion- more like a much-needed summary of the results as this was sometimes not clear in the result section itself. In addition, reproducibility is not given as material+method section is not complete but missing important information (and I only checked the sections where I have experience with).

The claim in the abstract (Line36f) that the results could define weakly/moderate neutralizing mAbs vs such with strong neutralization activity is in my opinion not met, as CoVA2-04 also binds the ACE2 binding motif (what "defines" a strong neutralizer)- but is only a moderate neutralizer. In general, the number of tested Ab is too low to conclude such a claim.

Line 56-57 "The principal target for development of effective antibody-based antiviral approaches is the viral Spike glycoprotein." And Line 69-71 "As the Spike protein serves as the first point-of-contact between the virus and the host, it represents a primary target for neutralizing antibodies against SARS-CoV-2 infection (14)." should be fused.

Line 92 It should be clearer that LSI-CoVA-014 up to 017 were discovered in this work. Please add ref 36 also to CoVA-02 (as LSI-CoVA-014 up to 017 don't have a reference it confuses) . The Material+Method section about the discovery of these Abs is quite short.

Line 100f: I am missing a theory about why antibody 4A8 is not/ weakly binding in ELISA but still moderately neutralize the virus? Maybe its due to different glycosylation as this Spike was produced in insect cells, but pseudovirus is derived from mammalian cells (just an idea as you didn't add where which material was used, its speculation from my side). It should be added to the discussion.

Line 143, Line 144 You state that "antibody-binding at RBD induces allosteric changes across the Spike trimer, resulting in its global destabilization that may lead to dissociation of adjacent monomers." Why not test this by SEC? Wouldn't you obtain Spike monomers + antibody (260kDa) vs Spike trimer + Ab with other mode of action ((~580/680/780) or extended data fig3)) c) in general I would like to see the SEC profile of your Spike protein as you did not use an artificial trimerization domain and I worry about quality/stability of the trimeric state of Spike during the experiments even if initially only trimeric peak was collected, as you might have an equilibrium of trimeric/monomeric Spike (what we observed in our experiments).

Line 210: Why is CoVA2-04 a weak neutralizer if you yourself claimed it to be a moderate neutralizer in fig 1c?

Line 239f This whole (and very interesting) paragraph does only have figures/tables/data in the extended data. You should think about restructuring the figures.

Line 268: Why not test all 9 Abs on the deglycosylated Spike?

Line 319 "hinge dynamics" you never used this term before or afterwards so I cannot see that you explored the possibility of it?

Line 324f: "We did not observe any enhancement in the neutralization efficacies of the two potent HuMAbs (CoVA2-39, 5A6) with NTD-binding LSI-CoVA-017" probably the resolution of your

method is not suitable to do so? You already detect nearly 100% using CoVA2-39 or 5A6 alone. How do you expect to see any additive effect?
Line 352 very interesting and conclusive result.
Line 363ff: Please try to restructure this paragraph its really difficult to follow for the reader.
Line375: is it surprising? Maybe its avidity effects?
Fig 1b) statistical analysis, how many experiments?
Fig 1c) number of experiments- standard deviation shown?
Figure 1d+e I don't get why this is not in one with figure 2a-c (its also described together in one section in the results part).
Figure 1 d) in the legend I and ii are against RBD in the figure its I and iii whereas ii is minus NTA
Figure 1 d) in this quality numbers on x axis are not readable
Fig.2) e+f are way smaller font than a,b,c, d
Fig.4) very clear figure that gives a good overview about the model
Extended Data Fig.1: what does the black half brackets under diagram b mean? Legend and y axis description should be added to every figure,.
Line504: If you worked with blood samples I would expect an ethical statement?
Line 518/519 what kind of branched Pei (e.g. Mw 600 or 70000), what company
Line 514f In general the transfection protocol is missing information like cell density, media, transfection volume, PEI : DNA ratio.
Line 514f As the glycosylation differs in Sf9 cells and HEK cells you should specify in the experiments if you used insect cell or HEK cell derived material-
Line 514ff: what about recombinant ACE2 -didn't you use that as well, how was it produced?
Line 539 please add information about Hexa Pro spike variant eg no furin site as the reader does have to look it up on addgene. Again it is unclear in what experiments you used what variant and why you chose the Hexapro variant.
Line 981 RBD (Wuhan)
Line 1014: I cannot find ACE2-binding inhibition assay in material and methods?
Line 1095 should be Table S3

Reviewer #2:

Remarks to the Author:

The study focuses on the characterisation of antibodies against SARS-CoV-2 sourced from convalescent COVID-19 patients. Based on the extent of neutralisation, the authors classify the antibodies into weak, moderate, or strong neutralisers followed by their epitope mapping using HDX-MS. In the next step, the mechanism of neutralisation upon binding to the viral spike protein is explored using HDX-MS for the three categories of neutralisers. The impact of mutations in the VoC and the associated conformational changes induced by antibody binding to the spike protein of different VoC are studied. Based on the results, the authors propose the use of antibody cocktails for the neutralisation of current and future VoC.

Overall, it is an extensive study detailing the conformational changes induced by the antibodies under study, upon binding to WT spike protein of SARS-CoV-2 and its VoC. It sheds light on the molecular mechanism of neutralisation of the virus by these antibodies and paves the way for the development of antibody-based therapies against SARS-CoV-2 VoC in the future.

Although the authors obtained a lot of data, but it is not conveyed in the best possible way to the reader. Most of the HDX figures are difficult to follow with the text and are not clear enough to convey the message.

An overall restructuring and rearrangement of the figures is required to go in line with the text. Very often the figures do not support the text e.g., line 121, 127-128, 129-130 (see below).

- The figures are not easy to understand and very often one must scroll through several figures to understand each result.
- For each result subsection, it would be good to have a separate figure especially for differential HDX results.
- The HDX figures need to be clearer and more detailed but less complex and at the same time easy to compare.
- Line 118 – Difficult to follow figures with text for e.g., text states extensive protection in CoVA-014, 015 and 016. But only CoVA16 is shown in figure 1, for CoVA 14 one needs to refer to the extended figure 1b and CoVA-015 HDX result is not there in these figures contrary to it being

mentioned in the text.

- Line 119 – Extended Fig1b, the primary sequence is missing on the figure so could not follow the protection in residues 361-395.
- Line 121 – The text refers to extended Fig1b for protection in CoVA-014, 015 and 016 but the figure shows only CoVA-014.
- Line 127-128 – The text refers to Ext figure 1d for CoVA-014, 015 and 016 but the figure shows CoVA2-39. Also increased exchange is not clear on Figure 1d.
- Line 129-130 – Increased destabilisation at ACE2 binding site (RBD) not clear from the figure.
- Line 130 – Figure 1e difficult to follow as the orientation of RBD (spike) not same and if rotated then no angle of rotation is stated.
- Line 133 – Difficult to read the residue numbers on Extended figure 2a.
- Line 136 – In fig 2ai, the arrow for peptide 166-182 does not point to the peptide showing increased uptake contrary to what is mentioned in the text.
- Line 139 – Residues 902-916 not visible on Fig2aii and difficult to follow on Extended Fig 2a.
- Line 157 – Fig2'B' is capital.
- Line 159 – The text mentions large scale deprotection but in the Figure 2b, the highlighted area shows mixed protection and deprotection. It would be better to label individual peptides to be in line with the text. Also, the uptake plots of individual peptides in insets can propagate the message better.
- Line 168 – Extended data?
- Line 176 – Figure reference?
- Line 182 – Central helix and S1/S2 cleavage site not labelled on Fig 2cii inset.
- Line 215 – Very difficult to compare the results when spread over multiple figures.
- Line 972 – Fig1di, add 'weak' neutralising to be consistent. Timepoints label should be on Fig1di or on all the three for better visibility.
- Line 974 – Better to mention 'isolated RBD or NTD' where applicable for clarity. Fig 1d and 1e should be a separate figure preferably with a separate legend for clarity. Also, because it is a separate subheading in the results.
- Line 977 – Fig1e should be numbered in relation to fig1d for clarity.
- Line 978 – Fig1ei and Fig1eiii seems to be the RBD but is labelled wrong in the legend.
- Line 1049 - Extended Fig 1b-1d, yellow and black boxes should be there on all 3.
- Methods: Line 667: Trapping time and gradient elution time?
- Line 679: Please detail the parameters used in DynamX to shortlist the peptides. Details of the software(s) used to show HDX data?

Reviewer #3:

Remarks to the Author:

Tulsian probes the role of allostery between various regions of the spike protein in SARS-CoV-2 variants. Data indicate some sort of allostery, but the results are very preliminary and needs further analysis and experiments. Testing the efficacy of antibodies in neutralizing SARS-CoV-2 variants has already been studied in more than 30+ publications, and I did not find any interesting. HDX-MS and other experimental data on allostery requires further detailed data analysis. Observing changes in HDX-MS rates does not prove the concerted motion of various regions in the spike protein. Authors need to show some serious analysis on calculating allosteric parameters such as change in Gibbs free energy of binding, which is quite standard now in the literature on protein allostery and cooperativity. Please refer to some of the new literature articles published in the past decade. Without such rigorous analysis, most conclusions are very hand wavy and do not strongly prove the concept of allostery. Other concerns include:

(1) Page 3, Line 93: CR3022 is originally identified in SARS-CoV patients and hence considered as a SARS-CoV antibody, not as SARS-CoV-2 antibody. Please correct the text.

(2) Page 11: Allostery can be one of the reasons underlying increased deuterium exchange, but this definitely needs further analysis. Without additional data analysis and experiments, authors do not provide solid support for allostery. HDX-MS analysis performed by authors are more like those studies published in 2000s, and not in 2020s.

(3) Page 13: Spike protein expressed in insect cells is not a good system to study, as it has been shown in the literature that glycosylation plays a big role on spike protein stability and function.

(4) Please show how pure your spike proteins are and their oligomerization status (PAGE, SEC-MALS, AUC, etc.). Impure spike proteins are known to oligomerize and generate multiple species which will lead to complicated results that can be interpreted wrongly.

(5) Page 16: Authors did not mention how they corrected for back exchange during their HDX-MS experiments. This requires adding control experiments to their HDX protocols. Increase in deuterium can be because of increased local breathing motions which are independent of allostery. Allostery involves subglobal unfolding and cooperative movement of various domains. The data analysis reported in the manuscript is very preliminary.

Response to Reviewers'

Reviewer #1:

Tulsian et al presents new data about how antibodies lead to binding- induced allosteric effects on the conformation of Spike and antibody. The main method is HDXMS to determine dynamics in binding. The authors tested nine mAbs, four of them were newly generated in this manuscript. First affinity+ kinetic and neutralization of these mAbs was tested and the mAbs were divided into weak, moderate and strong neutralizers. By HDXMS effect on RBD and or NTA-Spike of the binding – but only for six of the nine mAbs- was analysed and mapped on structures. Capture ELISA was performed to get combinational mAbs pairs. Some combinations were also tested in neutralization. Next binding to Wuhan, delta and Omicron of six of the nine mAbs was tested in ELISA, ACE2-binding inhibition assay and some of them in differenced in deuterium exchange. A model of how the different mAbs effect and neutralize was obtained. Glykan binding was also analysed but data can be only found in the supplement. As I am not in expert in structural biology (modelling) nor on HDXMS in I cannot comment on the quality of these data or on the material and method section.

In total, the manuscript comprises a lot of impressive data and experiments, interesting for the scientific community- BUT for some passages restructuring/rewriting (incl. reordering of figures) is advised as the structure/story is not clear and the reader is lost - this might also be caused by the density of data, which in general is a positive aspect but it should be appropriately packaged.

We thank the reviewer for their thorough appraisal of our manuscript. We have significantly restructured the text and the figures of the manuscript. The revised manuscript contains 9 Results subsections and 10 main figures. Also, we would like to clarify that the manuscript in the originally submitted form was transferred from *Nature* to *Nature Communications*, and was therefore organised as per the guidelines for Nature. We believe that the revised manuscript is now clearer and the data progression more logical.

Other major limitations are that some results remain unexplained (examples see below). Discussion in general is quite short and not really a discussion- more like a much-needed summary of the results as this was sometimes not clear in the result section itself.

We understand the reviewer's concerns and have amended and augmented the manuscript accordingly. Please see our point-by-point responses to the queries raised:

In addition, reproducibility is not given as material+method section is not complete but missing important information (and I only checked the sections where I have experience with).

We acknowledge this and have added all the necessary information in the Methods section for the readers to understand our experimental approaches, as well as providing the statistical significance used for the relevant Results sections.

The claim in the abstract (Line36f) that the results could define weakly/moderate neutralizing mAbs vs such with strong neutralization activity is in my opinion not met, as CoVA2-04 also binds the ACE2 binding motif (what “defines” a strong neutralizer)- but is only a moderate neutralizer. In general, the number of tested Ab is too low to conclude such a claim.

We thank the reviewer for highlighting this point. Yes, CoVA2-04 was indeed observed to be a moderate neutralizer that binds to the ACE2 binding motif on RBD. We have replaced ‘define’ with ‘distinguish’ in the revised abstract to prevent any misinterpretation.

Line 56-57 “The principal target for development of effective antibody-based antiviral approaches is the viral Spike glycoprotein.” And Line 69-71 “As the Spike protein serves as the first point-of-contact between the virus and the host, it represents a primary target for neutralizing antibodies against SARS-CoV-2 infection (14).” should be fused.

We would like to clarify that while the first statement was indeed part of the current version of the manuscript submitted, the second statement highlighted by the reviewer was part of a previous version of the manuscript deposited on bioRxiv. However, we have carefully reviewed and amended any duplicate sentences.

Line 92 It should be clearer that LSI-CoVA-014 up to 017 were discovered in this work. Please add ref 36 also to CoVA-02 (as LSI-CoVA-014 up to 017 don’t have a reference it confuses).

We thank the reviewer for highlighting this. We have clarified the antibodies discovered and characterized in this study and have cited necessary references for antibodies characterized previously.

The Material+Method section about the discovery of these Abs is quite short.

We have significantly augmented the revised manuscript to describe the discovery, expression, and purification of the antibodies used in this study.

Line 100f: I am missing a theory about why antibody 4A8 is not/ weakly binding in ELISA but still moderately neutralize the virus? Maybe its due to different glycosylation as this Spike was produced in insect cells, but pseudovirus is derived from mammalian cells (just an idea as you didn’t add where which material was used, its speculation from my side). It should be added to the discussion.

For neutralization and binding ELISA assays, we utilized a mammalian cell derived Spike HexaPro construct. Previous studies have characterized the epitopes and high-resolution structures of antibody 4A8 and its binding to Spike trimer, which were used to hypothesize it neutralizes by restraining the conformational changes of Spike upon binding. Additionally, our MD simulations and deglycosylation assays have shown that the glycan groups play an essential role in antigen recognition. For clarity, we have amended the Methods section to specify the origin of various isolated RBD and Spike constructs used in this study.

Line 143, Line 144 You state that “antibody-binding at RBD induces allosteric changes across the Spike trimer, resulting in its global destabilization that may lead to dissociation of adjacent monomers.” Why not test this by SEC? Wouldn't you obtain Spike monomers + antibody (260kDa) vs Spike trimer + Ab with other mode of action ((~580/680/780) or extended data fig3)) c) in general I would like to see the SEC profile of your Spike protein as you did not use an artificial trimerization domain and I worry about quality/stability of the trimeric state of Spike during the experiments even if initially only trimeric peak was collected, as you might have an equilibrium of trimeric/monomeric Spike (what we observed in our experiments).

We thank the reviewer for their insights. The purification and expression of the Spike protein was reported in our earlier referenced manuscript, wherein the size-exclusion chromatograms clearly showed Spike to be purified as a trimer. In this study, both insect and mammalian Spike constructs have a C-terminal trimerization domain. Indeed, the Spike trimer undergoes slow monomerization resulting in trimer-monomer heterogeneity. However, all our experiments were performed with freshly purified Spike trimer and with different aliquots to prevent freeze-thaw cycles.

For antibody binding effects, we have characterized the binding of LSI-CoVA-017 with Spike through size-exclusion chromatography and this data was included in the manuscript. As suggested by the reviewer, we performed additional analytical size-exclusion chromatography which shows the trimeric nature of Spike alone, and a left shift of the chromatographic peak in the presence of LSI-CoVA-016 and LSI-CoVA-017 antibodies indicates the formation of antibody-Spike complexes. The peak shifts were analyzed on denaturing gel electrophoresis to identify the peak composition. These images are shown below as a reference for reviewers. Binding of LSI-CoVA-016 to Spike led to large-scale increases in deuterium exchange, which reflects higher conformational dynamics upon antibody binding likely due to increased solvent accessibility. Hence, we hypothesize that the antibody-binding to RBD induces long-range allosteric changes and destabilizing effects.

Figure.

a. Size exclusion chromatography purification of hexapro Spike (b) Fractions from the middle peak of Spike were analyzed by negative staining electron microscopy. Chromatograms of purified Spike trimer (black trace) complexed with (c) LSI-CoVA-016 (red) and (d) LSI-CoVA-017 (blue) are shown. Corresponding chromatograms of free antibodies LSI-CoVA-016 (green) and LSI-CoVA-017 (pink) are also indicated as controls. (e) Peak fractions indicated with arrow were analyzed by reducing SDS PAGE analysis. Supplementary Fig. 5 also shows SEC chromatogram of hexapro Spike + LSI-CoVA-017.

Line 210: Why is CoVA2-04 a weak neutralizer if you yourself claimed it to be a moderate neutralizer in fig 1c?

CoVA2-04 is a moderate neutralizer. This error in the manuscript text has been corrected.

Line 239f This whole (and very interesting) paragraph does only have figures/tables/data in the extended data. You should think about restructuring the figures.

We thank the reviewer for this feedback. The section and figures have been restructured in the revised manuscript to highlight the importance of these results.

Line 268: Why not test all 9 Abs on the deglycosylated Spike?

We tested the effects of deglycosylation and glycosylation using isolated RBD and Spike constructs of Wuhan-Hu-1, Delta, and Omicron variants. This was part of the extended data in the earlier manuscript and in the revised manuscript, it is Supplementary Figure 10 with data in Supplementary Table S6.

Line 319 “hinge dynamics” you never used this term before or afterwards so I cannot see that you explored the possibility of it?

The term ‘hinge dynamics’ refers to the conformational changes observed at the region connecting the S1 and the S2 subunits, and has been edited in the revised manuscript. The earlier result sections reported the increased deuterium exchange at the various hotspots of the S2 subunit upon antibody binding. At this section, ‘hinge dynamics’ indicated the collective changes for the different antibodies.

Line 324f: “We did not observe any enhancement in the neutralization efficacies of the two potent HuMAbs (CoVA2-39, 5A6) with NTD-binding LSI-CoVA-017” probably the resolution of your method is not suitable to do so? You already detect nearly 100% using CoVA2-39 or 5A6 alone. How do you expect to see any additive effect?

We appreciate the reviewer’s concerns regarding resolution. We would like to point out that the strongly neutralizing antibodies CoVA2-39 and 5A6 alone showed 100% neutralization at the concentrations specified, with a possibility that the addition of another antibody may change the neutralization profile. For deeper understanding of the absolute *quantitative* comparison, additional experiments using a wider range of varying antibody concentrations, or an alternative method may be suited. In this study, we have highlighted the *qualitative* nature of the antibody competition.

Line 352 very interesting and conclusive result. Line 363ff: Please try to restructure this paragraph its really difficult to follow for the reader. Line375: is it surprising? Maybe its avidity effects? Fig 1b) statistical analysis, how many experiments? Fig 1c) number of experiments-standard deviation shown?

We have revised the manuscript to ensure these details are clearly expressed and have added the statistical parameters used. Binding activity measurements and neutralization assays were performed in triplicates, and for many points the standard deviations are too small to be visible. The raw data are tabulated as Source Data 1.

Figure 1d+e I don’t get why this is not in one with figure 2a-c (its also described together in one section in the results part). Figure 1 d) in the legend I and ii are against RBD in the figure

its I and iii whereas ii is minus NTA. Figure 1 d) in this quality numbers on x axis are not readable. Fig.2) e+f are way smaller font than a,b,c, d

The manuscript was directly transferred from Nature, with restriction in number of figures in the main manuscript. However, we have restructured the revised manuscript for clarity. The reviewer's suggestions to merge parts of different figures were also considered. The figures and legends in the revised manuscript now have appropriate font size for reading.

Fig.4) very clear figure that gives a good overview about the model.

We thank the reviewer for their appreciation and feedback.

Extended Data Fig.1: what does the black half brackets under diagram b mean? Legend and y axis description should be added to every figure.

The half brackets indicated that the peptides with overlapping residue numbers were sub-grouped to prevent text overlap. In the revised figures, we have indicated the residue numbers of each peptide, which correlates with the corresponding source data (or supplementary data) as indicated in the figure legends. Both x- and y- axes have been indicated for all figures in the revised manuscript.

Line504: If you worked with blood samples I would expect an ethical statement?

This work was conducted under an approved protocol from an Institutional Review Board. We have added the necessary statements to the ethical statement in the Methods section.

Line 518/519 what kind of branched Pei (e.g. Mw 600 or 70000), what company?

We used branched PEI stock solution (Sigma Cat. No. 408727) with an average Mw of ~25,000 by LS, and average Mw ~10,000 by GPC.

Line 514f - In general the transfection protocol is missing information like cell density, media, transfection volume, PEI : DNA ratio.

The revised manuscript describes the methods in significantly greater detail. Briefly, mammalian cells were cultured in freestyle f17 expression medium (Life Technologies). Cultures at a density of 1×10^6 cells per mL were transfected with 1mg of DNA per mL of culture at 1:3 ratio of DNA: branched PEI (Sigma-Aldrich). The culture supernatant was harvested for purification at day 6 post transfection.

Line 514f - As the glycosylation differs in Sf9 cells and HEK cells you should specify in the experiments if you used insect cell or HEK cell derived material.

In the revised manuscript, we have specified the nature of Spike constructs used for different experiments. Spike trimer purified from insect-cell culture was mainly used for HDX analysis, glycosylated vs deglycosylated Spike binding affinity to antibodies by QCM, while Spike HexaPro trimer purified from mammalian-cell culture was used for other binding analyses.

Line 514ff: what about recombinant ACE2 -didn't you use that as well, how was it produced?

Expression and purification of recombinant ACE2 was described in our previous manuscript – which has been cited: Raghuvamsi, P. V. et al. SARS-CoV-2 S protein:ACE2 interaction reveals novel allosteric targets. *eLife* **10** (2021).

Line 539 please add information about Hexa Pro spike variant e.g. no furin site as the reader does have to look it up on addgene. Again it is unclear in what experiments you used what variant and why you chose the Hexapro variant.

We thank the reviewer for highlighting this point. The revised manuscript has been modified to prevent any inconvenience or misunderstanding of the nature of the Spike construct used. This information regarding the Spike protein derived from the Wuhan-Hu-1 strain of virus was described in detail in our previous manuscript and cited accordingly. The details of the Spike constructs and the mutations have been added in the Methods section of the revised manuscript.

Line 981 RBD (Wuhan)

We have amended this in the revised manuscript.

Line 1014: I cannot find ACE2-binding inhibition assay in material and methods?

We thank the reviewer for pointing this out. The details for the ACE2-binding inhibition assay are now described in significant detail in the Methods section of the revised manuscript.

Line 1095 should be Table S3

We have amended this in the revised manuscript.

Reviewer #2 (Remarks to the Author):

The study focuses on the characterisation of antibodies against SARS-CoV-2 sourced from convalescent COVID-19 patients. Based on the extent of neutralisation, the authors classify the antibodies into weak, moderate, or strong neutralisers followed by their epitope mapping using HDX-MS. In the next step, the mechanism of neutralisation upon binding to the viral spike protein is explored using HDX-MS for the three categories of neutralisers. The impact of mutations in the VoC and the associated conformational changes induced by antibody binding to the spike protein of different VoC are studied. Based on the results, the authors propose the use of antibody cocktails for the neutralisation of current and future VoC. Overall, it is an extensive study detailing the conformational changes induced by the antibodies under study, upon binding to WT spike protein of SARS-CoV-2 and its VoC. It sheds light on the molecular mechanism of neutralisation of the virus by these antibodies and paves the way for the development of antibody-based therapies against SARS-CoV-2 VoC in the future. Although the authors obtained a lot of data, but it is not conveyed in the best possible way to the reader. Most of the HDX figures are difficult to follow with the text and are not clear enough to convey the message.

An overall restructuring and rearrangement of the figures is required to go in line with the text.

We thank the reviewer for their appraisal of our manuscript. We have taken their suggestions into consideration and have restructured the text and figures accordingly. We believe that the revised manuscript is now more logical and trust that it retains the readers' interests throughout.

Very often the figures do not support the text e.g., line 121, 127-128, 129-130 (see below).

- The figures are not easy to understand and very often one must scroll through several figures to understand each result.
- For each result subsection, it would be good to have a separate figure especially for differential HDX results.
- The HDX figures need to be clearer and more detailed but less complex and at the same time easy to compare.

We thank the reviewer for this feedback. We have made the necessary amendments to the figures in the revised manuscript.

- Line 118 – Difficult to follow figures with text for e.g., text states extensive protection in CoVA-014, 015 and 016. But only CoVA16 is shown in figure 1, for CoVA 14 one needs to refer to the extended figure 1b and CoVA-015 HDX result is not there in these figures contrary to it being mentioned in the text.

We acknowledge this reviewers' point. The manuscript was directly transferred from *Nature*, which had a restriction in the number of figures allowed in the main text. Hence, many of the figures were multi-panel with certain results shown in the supplementary information. However, we have considered the reviewer's suggestions and revised the figures as necessary.

In the revised manuscript, we have clearly illustrated the effects of each antibody on RBD or Spike proteins, with corresponding relevant information in Source and Supplementary files.

- Line 119 – Extended Fig1b, the primary sequence is missing on the figure so could not follow the protection in residues 361-395.
- Line 121 – The text refers to extended Fig1b for protection in CoVA-014, 015 and 016 but the figure shows only CoVA-014.

We have amended this in the revised manuscript.

- Line 127-128 – The text refers to Ext figure 1d for CoVA-014, 015 and 016 but the figure shows CoVA2-39. Also increased exchange is not clear on Figure 1d.

We have amended the figures in the revised manuscript, with different figures for each category of antibody.

- Line 129-130 – Increased destabilisation at ACE2 binding site (RBD) not clear from the figure.

We have amended this in the revised manuscript.

- Line 130 – Figure 1e difficult to follow as the orientation of RBD (spike) not same and if rotated then no angle of rotation is stated.

We have amended this in the revised manuscript with a similar orientation of the RBD, and shown it in surface representation.

- Line 133 – Difficult to read the residue numbers on Extended figure 2a.
- Line 136 – In fig 2ai, the arrow for peptide 166-182 does not point to the peptide showing increased uptake contrary to what is mentioned in the text.

We have amended this in the revised manuscript.

- Line 139 – Residues 902-916 not visible on Fig2aii and difficult to follow on Extended Fig 2a.

We have amended the figures for better clarity in the revised manuscript.

- Line 157 – Fig2'B' is capital.

We have amended this in the revised manuscript.

- Line 159 – The text mentions large scale deprotection but in the Figure 2b, the highlighted area shows mixed protection and deprotection. It would be better to label individual peptides

to be in line with the text. Also, the uptake plots of individual peptides in insets can propagate the message better.

We thank the reviewer for these suggestions. We have appropriately amended the figures in the revised manuscript. While the uptake plots of individual peptides could add the value to the figure, it may lead to figure cluttering. The raw data of deuterium exchange for all peptides and different conditions are tabulated as Source data and supplementary tables.

- Line 168 – Extended data?

We apologize for the missing text. The revised manuscript has all the details indicated at the appropriate places.

- Line 176 – Figure reference?

We appreciate the reviewer's concern. Comparative HDXMS analysis of LSI-CoVA-017 and 4A8 with isolated RBD was performed at 1- and 10-min labelling time points, and the data is tabulated as Source Data 2.

- Line 182 – Central helix and S1/S2 cleavage site not labelled on Fig 2cii inset.

We have amended this in the revised manuscript.

- Line 215 – Very difficult to compare the results when spread over multiple figures.

We have corrected this via multiple figures in the revised manuscript.

- Line 972 – Fig1di, add 'weak' neutralising to be consistent. Timepoints label should be on Fig1di or on all the three for better visibility.

We have amended this in the revised manuscript with individual figures for weakly- (Figure 2), moderately- (Figure 3) and strongly- (Figure 4) neutralizing antibodies.

- Line 974 – Better to mention 'isolated RBD or NTD' where applicable for clarity. Fig 1d and 1e should be a separate figure preferably with a separate legend for clarity. Also, because it is a separate subheading in the results.

We thank the reviewer for pointing this out. We have amended this in the revised manuscript.

- Line 977 – Fig1e should be numbered in relation to fig1d for clarity.
- Line 978 – Fig1ei and Fig1eiii seems to be the RBD but is labelled wrong in the legend.

We have amended this in the revised manuscript.

- Line 1049 - Extended Fig 1b-1d, yellow and black boxes should be there on all 3.

We appreciate the reviewer's feedback. We have amended this in the revised manuscript.

- Methods: Line 667: Trapping time and gradient elution time?

For HDXMS analysis, the trapping time was 3 min, and the actual gradient elution time was 7 min. These details have been added in the methods section (LCMS analysis) of the revised manuscript.

- Line 679: Please detail the parameters used in DynamX to shortlist the peptides. Details of the software(s) used to show HDX data?

The parameters used for filtering the peptides using DynamX are described previously and was cited. However, these parameters were consistent for all conditions and variants of isolated RBD and Spike constructs and details been added in the data processing section of the Methods section in the revised manuscript.

Reviewer #3 (Remarks to the Author):

Tulsian probes the role of allostery between various regions of the spike protein in SARS-CoV-2 variants. Data indicate some sort of allostery, but the results are very preliminary and needs further analysis and experiments. Testing the efficacy of antibodies in neutralizing SARS-CoV-2 variants has already been studied in more than 30+ publications, and I did not find any interesting. HDX-MS and other experimental data on allostery requires further detailed data analysis. Observing changes in HDX-MS rates does not prove the concerted motion of various regions in the spike protein. Authors need to show some serious analysis on calculating allosteric parameters such as change in Gibbs free energy of binding, which is quite standard now in the literature on protein allostery and cooperativity. Please refer to some of the new literature articles published in the past decade. Without such rigorous analysis, most conclusions are very hand wavy and do not strongly prove the concept of allostery.

We thank the reviewer for their comments and suggestions. We agree with the reviewer that testing the efficacy of antibodies alone has limited use in the post-vaccine state. Nonetheless, we would like to highlight the novel findings of the current work which contribute to our understanding of antibody responses to SARS-CoV-2

1) Most structural studies use Fab molecules to map the interface between antibody-antigen and report the corresponding changes in Spike. Here, we have used full-length IgGs binding to Spike, which is technologically more challenging but physiologically more relevant.

2) Our RBD-Fab modelling/simulations followed by ELISA and binding kinetics assays on deglycosylated Spike and IgGs have highlighted a novel glycan mediated- IgG binding mechanism. Interestingly, similar findings were recently reported for Ebola virus-Fab binding (Rayaprolu et.al *Cell, Host and Microbe* 2023, DOI: 10.1016/j.chom.2023.01.002).

3) New antibodies discovered in the current study add to the list of antibodies that retain binding to Spike Variants of Concern, and may be further rationally engineered for future purposes.

We agree with the reviewer's comment that a more rigorous analysis of deuterium uptake data is warranted to associate the IgG binding induced changes on Spike to allostery. As we have used full-length IgGs, other than allostery, avidity of IgG or destabilisation of trimeric quaternary contacts because of Spike cross-linking may contribute to the observed changes deuterium uptake of Spike-IgG states. Therefore, we have revised the main text and elaborated on plausible reasons for changes in deuterium uptake upon IgG binding to Spike.

We would like to highlight that the analysis and interpretation were done as comparison between bound- and free- states of the respective constructs. Any significant changes in deuterium exchange at local or distal sites upon formation of a complex with antibody were reported. Multiple HDXMS studies and literature support this concept of orthostery and allostery. We would also like to point out that previously published literature on SARS-CoV-2 Spike protein with ACE2 and antibodies have primarily focused on identifying the binding sites, rather than looking at the overall conformational dynamics. Papers using HDXMS to understand conformational dynamics of SARS-CoV-2 have interpreted the data similar to that

shown in our paper. Some of these studies include Calvaresi V. *et.al.* (*Nat Commun* 2023, DOI: 10.1038/s41467-023-36745-0), Costello S.M. *et. al.* (*Nat Struct Mol Biol* 2022, DOI: 10.1038/s41594-022-00735-5), Braet S.M. *et. al.* (*eLife* 2023, DOI: 10.7554/eLife.82584).

As per the reviewer's suggestions, we have performed additional analyses and our detailed responses to the respective queries are listed below.

Other concerns include:

(1) Page 3, Line 93: CR3022 is originally identified in SARS-CoV patients and hence considered as a SARS-CoV antibody, not as SARS-CoV-2 antibody. Please correct the text.

We have amended this in the revised manuscript.

(2) Page 11: Allostery can be one of the reasons underlying increased deuterium exchange, but this definitely needs further analysis. Without additional data analysis and experiments, authors do not provide solid support for allostery. HDX-MS analysis performed by authors are more like those studies published in 2000s, and not in 2020s.

We understand the reviewer's concerns. However, we would like to highlight that the analysis and interpretation were done as a comparison between bound- and free- states of the respective constructs. Any significant changes in deuterium exchange at local or distal sites upon formation of a complex with antibody were reported. Many HDXMS studies and literature support this concept of orthostery and allostery. We would also like to point out that previously published literature on SARS-CoV-2 Spike protein with ACE2 and antibodies have primarily focused on identifying the binding sites, rather than looking at the overall conformational dynamics.

We agree with the reviewer's comment that a more rigorous analysis of deuterium uptake data is warranted to associate the IgG binding induced changes on Spike to allostery. As we are using full length IgGs, other than allostery, avidity of IgG or destabilisation of trimeric quaternary contacts because of Spike cross-linking may contribute to the observed changes deuterium uptake in Spike-IgG states. Therefore, we revised the main text and have elaborated on plausible reasons for changes in deuterium uptake upon IgG binding to Spike.

We would like to highlight that we are extending the current work to verify allosteric causality of IgG binding to Spike by using PyHDX module to analyse our deuterium data comprising of nine IgG antibodies with Spike Wuhan-Hu-1 and variant states. We present here some preliminary analysis we performed on our HDXMS data using the PyHDX (Smit J.H et.al 2021) module. PyHDX predicts residue level protection factors by using a Lagrangian function with the stochastic gradient descent method within a machine learning framework. Protection factors (PFs) of a peptide or residue is the ratio of deuterium exchange rate of the peptide or residue in open (deuterium competent state) and closed (deuterium exchange incompetent) states. The PF is a measure of ease of deuterium exchange and relates to the difference in Gibbs free energy between the closed and the open state, wherein large PF corresponds to large ΔG

and a higher propensity of the peptide to be stable. The plot below shows the $\Delta\Delta G$ of calculated from deuterium exchange differences between Spike bound to IgG and free Spike protein.

$$PF = e^{\Delta G/RT}$$

As optimizing and implementing PyHDX coupled with other orthogonal analysis tools to quantify allostery, including Allosigma (Guarnera and Berezovsky 2016b, and Guarnera and Berezovsky 2019b), adds significant amount of data to the current data-dense manuscript. We are reporting deuterium exchange difference and plausible explanations for differential uptake in the presence of IgGs as described above. These results are being prepared as a follow-up manuscript using PyHDX and other tools to quantify allostery. Some of the preliminary results from our analysis of Spike-IgG complexes using PyHDX are shown below:

Figure: Monitoring effects of antibody binding to Spike protein using PyHDX.

Difference plots showing the predicted difference in free energy at an amino acid resolution between Spike: IgG complex and free Spike states using PyHDX module. Positive (blue) and negative (green) differences in $\Delta\Delta G$ indicate stabilisation or destabilisation of Spike in the

presence of IgG respectively. Comparing Spike:CoVA39 complex (top) with other Spike:LSI-CoVA-014 and Spike:LSI-CoVA-016 shows that receptor binding motif binding IgG (CoVA2-39) stabilizes Spike to a greater extent than the cryptic site binding IgGs, which supports our proposed mechanism. It is noteworthy that high values of regularizers were chosen in PyHDX, in this case $\lambda_1=0.5$, $\lambda_2=1.0$, to suppress artefactual $\Delta\Delta G$ due to lack of overlapping peptides or time points in the HDXMS data. Residues showing no significant differences are highlighted in white and covariance is shown as error bars.

(3) Page 13: Spike protein expressed in insect cells is not a good system to study, as it has been shown in the literature that glycosylation plays a big role on spike protein stability and function.

We understand the reviewer's concerns about the role of glycosylation in Spike stability and function. We showed that our insect-cell expressed Spike forms a stable trimer, similar to HexaPro Spike from mammalian cells and is glycosylated. While the specific glycan composition varies on the protein expression system used and may have on specific interactions, the initial glycan moieties binding to the residues tend to be similar. These were reflected in our MD simulations results. Nonetheless, our biophysical techniques show that our purified Spike protein is trimeric and folded, as shown in the figures below.

(4) Please show how pure your spike proteins are and their oligomerization status (PAGE, SEC-MALS, AUC, etc.). Impure spike proteins are known to oligomerize and generate multiple species which will lead to complicated results that can be interpreted wrongly.

Our analytical size-exclusion chromatography (SEC-MALS) and preliminary negative stain electron microscopy clearly indicates that the Spike protein was pure and homogenous. The purity, homogeneity, and the oligomerization nature of Spike protein and other variants are highlighted in the figures below. The purification profiles of various RBD and Spike constructs determined by SDS-PAGE analysis has been added as Supplementary Figure 17 in the revised manuscript.

Figure. Homogeneity of Spike (Wuhan) trimer.

Molecular weight analysis of insect-cell expressed Spike by size exclusion chromatography – multiple angle light scattering (SEC-MALS). The measured molecular weight (kDa) is 482 ($\pm 1.7\%$).

(5) Page 16: Authors did not mention how they corrected for back exchange during their HDX-MS experiments. This requires adding control experiments to their HDX protocols. Increase in deuterium can be because of increased local breathing motions which are independent of allostery. Allostery involves subglobal unfolding and cooperative movement of various domains. The data analysis reported in the manuscript is very preliminary.

We acknowledge the reviewer's point regarding back-exchange correction. We would like to highlight that all the analysis and interpretation of HDXMS in this study is based on comparative analysis (difference plots) of Spike or RBD in the presence and absence of antibody or ACE2 protein. As a comparative analysis was performed, back-exchange correction of deuterium exchange values would be cancelled out in both conditions. This point was also highlighted in the HDXMS-community paper. However, the intrinsic dynamics of Spike(Wuhan-Hu-1) and isolated RBD(Wuhan-Hu-1) were probed with maximally-deuterated states and this was described in our previous study (Raghuvamsi et. al, eLife, 2021).

As highlighted by the reviewer, increased deuterium exchange could be because of local breathing motions or sub-global unfolding events or cooperative movement of various domains. These events were incurred upon binding of antibodies to Spike and isolated RBD proteins, as was observed for nine different antibodies. Most experiments were carried out as biological replicates each with technical triplicates, and their average values were used for analysis. Due to extensive amounts of data, the HDXMS analysis was restricted to simple observations.

Reviewers' Comments:

Reviewer #1:

Remarks to the Author:

The manuscript has been improved. A major point is that I really doubt the quality of the Spike protein is good enough to get conclusive results (figure for reviewer 1). Also I still find the discussion quite weak.

Response to Reviewers'

Reviewer #1:

Tulsian et al presents new data about how antibodies lead to binding- induced allosteric effects on the conformation of Spike and antibody. The main method is HDXMS to determine dynamics in binding. The authors tested nine mAbs, four of them were newly generated in this manuscript. First affinity+ kinetic and neutralization of these mAbs was tested and the mAbs were divided into weak, moderate and strong neutralizers. By HDXMS effect on RBD and or NTA-Spike of the binding – but only for six of the nine mAbs- was analysed and mapped on structures. Capture ELISA was performed to get combinational mAbs pairs. Some combinations were also tested in neutralization. Next binding to Wuhan, delta and Omicron of six of the nine mAbs was tested in ELISA, ACE2-binding inhibition assay and some of them in differenced in deuterium exchange. A model of how the different mAbs effect and neutralize was obtained. Glykan binding was also analysed but data can be only found in the supplement.

As I am not in expert in structural biology (modelling) nor on HDXMS in I cannot comment on the quality of these data or on the material and method section.

In total, the manuscript comprises a lot of impressive data and experiments, interesting for the scientific community- BUT for some passages restructuring/rewriting (incl. reordering of figures) is advised as the structure/story is not clear and the reader is lost - this might also be caused by the density of data, which in general is a positive aspect but it should be appropriately packaged.

We thank the reviewer for their thorough appraisal of our manuscript. We have significantly restructured the text and the figures of the manuscript. The revised manuscript contains 9 Results subsections and 10 main figures. Also, we would like to clarify that the manuscript in the originally submitted form was transferred from Nature to Nature Communications, and was therefore organised as per the guidelines for Nature. We believe that the revised manuscript is now clearer and the data progression more logical.

The revised manuscript is indeed clearer, still (due to many data) not easy to follow.

Other major limitations are that some results remain unexplained (examples see below). Discussion in general is quite short and not really a discussion- more like a much-needed summary of the results as this was sometimes not clear in the result section itself.

We understand the reviewer's concerns and have amended and augmented the manuscript accordingly. Please see our point-by-point responses to the queries raised:

I cannot agree that the discussion is much improved. It still reads like a summary and the real discussion of the results is short or totally absent.

In addition, reproducibility is not given as material+method section is not complete but missing important information (and I only checked the sections where I have experience with).

We acknowledge this and have added all the necessary information in the Methods section for the readers to understand our experimental approaches, as well as providing the statistical significance used for the relevant Results sections.

The Method section is much improved and complete from my point of view.

The claim in the abstract (Line36f) that the results could define weakly/moderate neutralizing mAbs vs such with strong neutralization activity is in my opinion not met, as CoVA2-04 also binds the ACE2 binding motif (what "defines" a strong neutralizer)- but is only a moderate neutralizer. In general, the number of tested Ab is too low to conclude such a claim.

We thank the reviewer for highlighting this point. Yes, CoVA2-04 was indeed observed to be a moderate neutralizer that binds to the ACE2 binding motif on RBD. We have replaced 'define' with 'distinguish' in the revised abstract to prevent any misinterpretation.

I don't agree that a simple change from "define" to "distinguish" solves the problem. When this classification you want to "define/distinguish" does not even work for one of six antibodies tested, I am in doubt that the conclusion is reasonable. That's by the way a good point for the still missing discussion.

Line 56-57 "The principal target for development of effective antibody-based antiviral approaches is the viral Spike glycoprotein." And Line 69-71 "As the Spike protein serves as the first point-of-contact between the virus and the host, it represents a primary target for neutralizing antibodies against SARS-CoV-2 infection (14)." should be fused.

We would like to clarify that while the first statement was indeed part of the current version of the manuscript submitted, the second statement highlighted by the reviewer was part of a previous version of the manuscript deposited on bioRxiv. However, we have carefully reviewed and amended any duplicate sentences.

Ok in the revised manuscript

Line 92 It should be clearer that LSI-CoVA-014 up to 017 were discovered in this work. Please add ref 36 also to CoVA-02 (as LSI-CoVA-014 up to 017 don't have a reference it confuses).

We thank the reviewer for highlighting this. We have clarified the antibodies discovered and characterized in this study and have cited necessary references for antibodies characterized previously.

This part is clearer now.

The Material+Method section about the discovery of these Abs is quite short.

We have significantly augmented the revised manuscript to describe the discovery, expression, and purification of the antibodies used in this study.

Yes, this part is much improved and sufficient

Line 100f: I am missing a theory about why antibody 4A8 is not/ weakly binding in ELISA but still moderately neutralize the virus? Maybe its due to different glycosylation as this Spike was produced in insect cells, but pseudovirus is derived from mammalian cells (just an idea as you didn't add where which material was used, its speculation from my side). It should be added to the discussion.

For neutralization and binding ELISA assays, we utilized a mammalian cell derived Spike HexaPro construct. Previous studies have characterized the epitopes and high-resolution structures of antibody 4A8 and its binding to Spike trimer, which were used to hypothesize it neutralizes by restraining the conformational changes of Spike upon binding. Additionally, our MD simulations and deglycosylation assays have shown that the glycan groups play an essential role in antigen recognition. For clarity, we have amended the Methods section to specify the origin of various isolated RBD and Spike constructs used in this study.

Thank you for this explanation and the clarification in the method part – yet, this explanation also belongs to the discussion part.

Line 143, Line 144 You state that "antibody-binding at RBD induces allosteric changes across the Spike trimer, resulting in its global destabilization that may lead to dissociation of adjacent monomers." Why not test this by SEC? Wouldn't you obtain Spike monomers + antibody (260kDa) vs Spike trimer + Ab with other mode of action ((~580/680/780) or extended data fig3)) c) in general I would like to see the SEC profile of your Spike protein as you did not use an artificial trimerization domain and I worry about quality/stability of the trimeric state of Spike during the experiments even if initially only trimeric peak was collected, as you might have an equilibrium of trimeric/monomeric Spike (what we observed in our experiments).

We thank the reviewer for their insights. The purification and expression of the Spike protein was reported in our earlier referenced manuscript, wherein the size-exclusion chromatograms clearly showed Spike to be purified as a trimer. In this study, both insect and mammalian Spike constructs have a C-terminal trimerization domain. Indeed, the Spike trimer undergoes slow monomerization resulting in trimer-monomer heterogeneity. However, all our experiments were performed with freshly purified Spike trimer and with different aliquots to prevent freeze-thaw cycles.

For antibody binding effects, we have characterized the binding of LSI-CoVA-017 with Spike through size-exclusion chromatography and this data was included in the manuscript. As suggested by the reviewer, we performed additional analytical size-exclusion chromatography which shows the trimeric nature of Spike alone, and a left shift of the chromatographic peak in the presence of LSI-CoVA-016 and LSI-CoVA-017 antibodies indicates the formation of antibody-Spike complexes. The peak shifts were analyzed on denaturing gel electrophoresis to identify the peak composition. These images are shown below as a reference for reviewers. Binding of LSI-CoVA-016 to Spike led to large-scale increases in deuterium exchange, which reflects higher conformational dynamics upon antibody binding likely due to increased solvent accessibility. Hence, we hypothesize that the antibody-binding to RBD induces long-range allosteric changes and destabilizing effects.

Figure.

a. Size exclusion chromatography purification of hexaprop Spike (b) Fractions from the middle peak of Spike were analyzed by negative staining electron microscopy. Chromatograms of purified Spike trimer (black trace) complexed with (c) LSI-CoVA-016 (red) and (d) LSI-CoVA-017 (blue) are shown. Corresponding chromatograms of free antibodies LSI-CoVA-016 (green) and LSI-CoVA-017 (pink) are also indicated as controls. (e) Peak fractions indicated with arrow were analyzed by reducing SDS PAGE analysis. Supplementary Fig. 5 also shows SEC chromatogram of hexaprop Spike + LSI-CoVA-017.

I thank the reviewers for their figures as reference for the reviewers. I assume that the SEC column used here was the same as in the manuscript: Superose 61535 Increase 10/300 GL gel filtration column? The profile of your Spike trimer looks not as expected- is that really your purified Spike trimer used in the experiments? In that case you have a lot of aggregates and quite a reasonable amount of Monomers.... I guess max 50% of the total "purified trimer" protein is indeed the trimer!

no matter if that's fresh and not frozen Spike: The quality of this Spike protein is really BAD!!!! And I now question all the experiments performed with such material!

Line 210: Why is CoVA2-04 a weak neutralizer if you yourself claimed it to be a moderate neutralizer in fig 1c?

CoVA2-04 is a moderate neutralizer. This error in the manuscript text has been corrected.

ok.

Line 239f This whole (and very interesting) paragraph does only have figures/tables/data in the extended data. You should think about restructuring the figures.

We thank the reviewer for this feedback. The section and figures have been restructured in the revised manuscript to highlight the importance of these results.

The structure is better.

Line 268: Why not test all 9 Abs on the deglycosylated Spike?

We tested the effects of deglycosylation and glycosylation using isolated RBD and Spike constructs of Wuhan-Hu-1, Delta, and Omicron variants. This was part of the extended data in the earlier manuscript and in the revised manuscript, it is Supplementary Figure 10 with data in Supplementary Table S6.

ok.

Line 319 "hinge dynamics" you never used this term before or afterwards so I cannot see that you explored the possibility of it?

The term 'hinge dynamics' refers to the conformational changes observed at the region connecting the S1 and the S2 subunits, and has been edited in the revised manuscript. The earlier result sections reported the increased deuterium exchange at the various hotspots of the S2 subunit upon antibody binding. At this section, 'hinge dynamics' indicated the collective changes for the different antibodies.

is ok.

Line 324f: "We did not observe any enhancement in the neutralization efficacies of the two potent HuMAbs (CoVA2-39, 5A6) with NTD-binding LSI-CoVA-017" probably the resolution of your method is not suitable to do so? You already detect nearly 100% using CoVA2-39 or 5A6 alone. How do you expect to see any additive effect?

We appreciate the reviewer's concerns regarding resolution. We would like to point out that the strongly neutralizing antibodies CoVA2-39 and 5A6 alone showed 100% neutralization at the concentrations specified, with a possibility that the addition of another antibody may change the neutralization profile. For deeper understanding of the absolute quantitative comparison, additional experiments using a wider range of varying antibody concentrations, or an alternative method may be suited. In this study, we have highlighted the qualitative nature of the antibody competition.

is ok with the addition of "This could be possibly due to the limitation of the method and warrants alternative method for quantification purposes."

Line 352 very interesting and conclusive result. Line 363ff: Please try to restructure this paragraph its really difficult to follow for the reader. Line375: is it surprising? Maybe its avidity effects? Fig 1b) statistical analysis, how many experiments? Fig 1c) number of experiments- standard deviation shown?

We have revised the manuscript to ensure these details are clearly expressed and have added the statistical parameters used. Binding activity measurements and neutralization assays were performed in triplicates, and for many points the standard deviations are too small to be visible. The raw data are tabulated as Source Data 1.

is ok.

Figure 1d+e I don't get why this is not in one with figure 2a-c (its also described together in one section in the results part). Figure 1 d) in the legend I and ii are against RBD in the figure its I and iii whereas ii is minus NTA. Figure 1 d) in this quality numbers on x axis are not readable. Fig.2) e+f are way smaller font than a,b,c, d

The manuscript was directly transferred from Nature, with restriction in number of figures in the

main manuscript. However, we have restructured the revised manuscript for clarity. The reviewer's suggestions to merge parts of different figures were also considered. The figures and legends in the revised manuscript now have appropriate font size for reading.

The arrangement of the figures is improved.

Fig.4) very clear figure that gives a good overview about the model.

We thank the reviewer for their appreciation and feedback.

Extended Data Fig.1: what does the black half brackets under diagram b mean? Legend and y axis description should be added to every figure.

The half brackets indicated that the peptides with overlapping residue numbers were sub-grouped to prevent text overlap. In the revised figures, we have indicated the residue numbers of each peptide, which correlates with the corresponding source data (or supplementary data) as indicated in the figure legends. Both x- and y- axes have been indicated for all figures in the revised manuscript.

Is okay – just add also a unit to x-axis of Sup. Fig8 c

Line504: If you worked with blood samples I would expect an ethical statement?

This work was conducted under an approved protocol from an Institutional Review Board. We have added the necessary statements to the ethical statement in the Methods section.

ok

Line 518/519 what kind of branched PEI (e.g. Mw 600 or 70000), what company?

We used branched PEI stock solution (Sigma Cat. No. 408727) with an average Mw of ~25,000 by LS, and average Mw ~10,000 by GPC.

ok.

Line 514f - In general the transfection protocol is missing information like cell density, media, transfection volume, PEI : DNA ratio.

The revised manuscript describes the methods in significantly greater detail. Briefly, mammalian cells were cultured in freestyle f17 expression medium (Life Technologies). Cultures at a density of 1×10^6 cells per mL were transfected with 1mg of DNA per mL of culture at 1:3 ratio of DNA: branched PEI (Sigma-Aldrich). The culture supernatant was harvested for purification at day 6 post transfection.

ok.

Line 514f - As the glycosylation differs in Sf9 cells and HEK cells you should specify in the experiments if you used insect cell or HEK cell derived material.

In the revised manuscript, we have specified the nature of Spike constructs used for different experiments. Spike trimer purified from insect-cell culture was mainly used for HDX analysis, glycosylated vs deglycosylated Spike binding affinity to antibodies by QCM, while Spike HexaPro trimer purified from mammalian-cell culture was used for other binding analyses.

It is still not clear to me in the revised manuscript where you used what material where did you put that information? – e.g. Figure 1 was it insect cell derived or mammalian ? - I only found the term "insect" 2x in the manuscript... –You should at least add this information in the material section or in the description of each figure.

Line 514ff: what about recombinant ACE2 -didn't you use that as well, how was it produced?

Expression and purification of recombinant ACE2 was described in our previous manuscript – which has been cited: Raghuvamsi, P. V. et al. SARS-CoV-2 S protein:ACE2 interaction reveals novel allosteric targets. eLife 10 (2021).

Please mention this in the manuscript and cite the paper e.g. in the method section for ACE2 production.

Line 539 please add information about Hexa Pro spike variant e.g. no furin site as the reader does have to look it up on addgene. Again it is unclear in what experiments you used what variant and why you chose the Hexapro variant.

We thank the reviewer for highlighting this point. The revised manuscript has been modified to prevent any inconvenience or misunderstanding of the nature of the Spike construct used. This information regarding the Spike protein derived from the Wuhan-Hu-1 strain of virus was described in detail in our previous manuscript and cited accordingly. The details of the Spike constructs and the mutations have been added in the Methods section of the revised manuscript.

ok.

Line 981 RBD (Wuhan)

We have amended this in the revised manuscript.

ok

Line 1014: I cannot find ACE2-binding inhibition assay in material and methods?

We thank the reviewer for pointing this out. The details for the ACE2-binding inhibition assay are now described in significant detail in the Methods section of the revised manuscript.

ok

Line 1095 should be Table S3

We have amended this in the revised manuscript.

ok.

Reviewer #2:

Remarks to the Author:

The authors have adequately responded to the comments and criticism.

Response to Reviewers'

Reviewer #1 (Remarks to the Author - see attached PDF for a formatted version with figures):

The manuscript has been improved. A major point is that I really doubt the quality of the Spike protein is good enough to get conclusive results (figure for reviewer 1). Also I still find the discussion quite weak.

Reviewer #1:

Tulsian et al presents new data about how antibodies lead to binding- induced allosteric effects on the conformation of Spike and antibody. The main method is HDXMS to determine dynamics in binding. The authors tested nine mAbs, four of them were newly generated in this manuscript. First affinity+ kinetic and neutralization of these mAbs was tested and the mAbs were divided into weak, moderate and strong neutralizers. By HDXMS effect on RBD and or NTA-Spike of the binding – but only for six of the nine mAbs- was analysed and mapped on structures. Capture ELISA was performed to get combinational mAbs pairs. Some combinations were also tested in neutralization. Next binding to Wuhan, delta and Omicron of six of the nine mAbs was tested in ELISA, ACE2-binding inhibition assay and some of them in differenced in deuterium exchange. A model of how the different mAbs effect and neutralize was obtained. Glykan binding was also analysed but data can be only found in the supplement.

As I am not in expert in structural biology (modelling) nor on HDXMS in I cannot comment on the quality of these data or on the material and method section.

Query1: In total, the manuscript comprises a lot of impressive data and experiments, interesting for the scientific community- BUT for some passages restructuring/rewriting (incl. reordering of figures) is advised as the structure/story is not clear and the reader is lost - this might also be caused by the density of data, which in general is a positive aspect but it should be appropriately packaged.

Response1: We thank the reviewer for their thorough appraisal of our manuscript. We have significantly restructured the text and the figures of the manuscript. The revised manuscript contains 9 Results subsections and 10 main figures. Also, we would like to clarify that the manuscript in the originally submitted form was transferred from Nature to Nature Communications, and was therefore organised as per the guidelines for Nature. We believe that the revised manuscript is now clearer and the data progression more logical.

Query 1A: The revised manuscript is indeed clearer, still (due to many data) not easy to follow.

Query 2: Other major limitations are that some results remain unexplained (examples see below). Discussion in general is quite short and not really a discussion- more like a much-needed summary of the results as this was sometimes not clear in the result section itself.

Response 2: We understand the reviewer's concerns and have amended and augmented the manuscript accordingly. Please see our point-by-point responses to the queries raised:

Query 2A: I cannot agree that the discussion is much improved. It still reads like a summary and the real discussion of the results is short or totally absent.

Response 2A: We thank the reviewer for their further insights to improve the manuscript. In the revised manuscript, we have significantly amended the discussion for better understanding and readability.

Query 3: In addition, reproducibility is not given as material+method section is not complete but

missing important information (and I only checked the sections where I have experience with).

Response 3: We acknowledge this and have added all the necessary information in the Methods section for the readers to understand our experimental approaches, as well as providing the statistical significance used for the relevant Results sections.

The Method section is much improved and complete from my point of view.

Query 4: The claim in the abstract (Line36f) that the results could define weakly/moderate neutralizing mAbs vs such with strong neutralization activity is in my opinion not met, as CoVA2-04 also binds the ACE2 binding motif (what “defines” a strong neutralizer)- but is only a moderate neutralizer. In general, the number of tested Ab is too low to conclude such a claim.

Response 4: We thank the reviewer for highlighting this point. Yes, CoVA2-04 was indeed observed to be a moderate neutralizer that binds to the ACE2 binding motif on RBD. We have replaced ‘define’ with ‘distinguish’ in the revised abstract to prevent any misinterpretation.

Query 4A: I don’t agree that a simple change from “define” to “distinguish” solves the problem. When this classification you want to “define/distinguish” does not even work for one of six antibodies tested, I am in doubt that the conclusion is reasonable. That’s by the way a good point for the still missing discussion.

Response 4A: The 9 different antibodies tested were classified into weak, moderate, and strong neutralizers based on their neutralization efficacies and their corresponding epitope sites on Spike protein. The conformational dynamics elicited by different antibodies in each of these three categories are distinct from each other. The ‘weak’ antibodies have a mild effect on the dynamics of the Spike trimer, inducing destabilization/increased dynamics. On the other hand, the strong neutralizing antibodies tested showed large-scale conformational rigidity of the Spike trimer. The moderate neutralizers showed intermediate effects of conformational changes on the Spike trimer. Accordingly, these three distinct effects complemented with neutralization efficacies were used to classify and distinguish the antibodies. This inference has also been added in the revised discussion (paragraphs 3 and 4 of discussion).

Query 5: Line 56-57 “The principal target for development of effective antibody-based antiviral approaches is the viral Spike glycoprotein.” And Line 69-71 “As the Spike protein serves as the first point-of-contact between the virus and the host, it represents a primary target for neutralizing antibodies against SARS-CoV-2 infection (14).” should be fused.

Response 5: We would like to clarify that while the first statement was indeed part of the current version of the manuscript submitted, the second statement highlighted by the reviewer was part of a previous version of the manuscript deposited on bioRxiv. However, we have carefully reviewed and amended any duplicate sentences.

Ok in the revised manuscript

Query 6: Line 92 It should be clearer that LSI-CoVA-014 up to 017 were discovered in this work. Please add ref 36 also to CoVA-02 (as LSI-CoVA-014 up to 017 don’t have a reference it confuses).

Response 6: We thank the reviewer for highlighting this. We have clarified the antibodies discovered and characterized in this study and have cited necessary references for antibodies characterized previously.

This part is clearer now.

Query 7: The Material+Method section about the discovery of these Abs is quite short.

Response 7: We have significantly augmented the revised manuscript to describe the discovery, expression, and purification of the antibodies used in this study.

Yes, this part is much improved and sufficient

Query 8: Line 100f: I am missing a theory about why antibody 4A8 is not/ weakly binding in ELISA but still moderately neutralize the virus? Maybe its due to different glycosylation as this Spike was produced in insect cells, but pseudovirus is derived from mammalian cells (just an idea as you didn't add where which material was used, its speculation from my side). It should be added to the discussion.

Response 8: For neutralization and binding ELISA assays, we utilized a mammalian cell derived Spike HexaPro construct. Previous studies have characterized the epitopes and high-resolution structures of antibody 4A8 and its binding to Spike trimer, which were used to hypothesize it neutralizes by restraining the conformational changes of Spike upon binding. Additionally, our MD simulations and deglycosylation assays have shown that the glycan groups play an essential role in antigen recognition. For clarity, we have amended the Methods section to specify the origin of various isolated RBD and Spike constructs used in this study.

Thank you for this explanation and the clarification in the method part – yet, this explanation also belongs to the discussion part.

We have edited this paragraph and moved some sentences to the discussion section in the revised manuscript.

Query 9: Line 143, Line 144 You state that “antibody-binding at RBD induces allosteric changes across the Spike trimer, resulting in its global destabilization that may lead to dissociation of adjacent monomers.” Why not test this by SEC? Wouldn't you obtain Spike monomers + antibody (260kDa) vs Spike trimer + Ab with other mode of action ((~580/680/780) or extended data fig3)) c) in general I would like to see the SEC profile of your Spike protein as you did not use an artificial trimerization domain and I worry about quality/stability of the trimeric state of Spike during the experiments even if initially only trimeric peak was collected, as you might have an equilibrium of trimeric/monomeric Spike (what we observed in our experiments).

Response 9: We thank the reviewer for their insights. The purification and expression of the Spike protein was reported in our earlier referenced manuscript, wherein the size-exclusion chromatograms clearly showed Spike to be purified as a trimer. In this study, both insect and mammalian Spike constructs have a C-terminal trimerization domain. Indeed, the Spike trimer undergoes slow monomerization resulting in trimer-monomer heterogeneity. However, all our experiments were performed with freshly purified Spike trimer and with different aliquots to prevent freeze-thaw cycles. For antibody binding effects, we have characterized the binding of LSI-CoVA-017 with Spike through size-exclusion chromatography and this data was included in the manuscript. As suggested by the reviewer, we performed additional analytical size-exclusion chromatography which shows the trimeric nature of Spike alone, and a left shift of the chromatographic peak in the presence of LSI-CoVA-016 and LSI-CoVA-017 antibodies indicates the formation of antibody-Spike complexes. The peak shifts were analyzed on denaturing gel electrophoresis to identify the peak composition. These images are shown below as a reference for reviewers. Binding of LSI-CoVA-016 to Spike led to large-scale increases in deuterium exchange, which reflects higher conformational dynamics upon antibody binding likely due to increased solvent accessibility. Hence, we hypothesize that the antibody-binding

to RBD induces long-range allosteric changes and destabilizing effects.

Figure.

a. Size exclusion chromatography purification of hexaproteins Spike (b) Fractions from the middle peak of Spike were analyzed by negative staining electron microscopy. Chromatograms of purified Spike trimer (black trace) complexed with (c) LSI-CoVA-016 (red) and (d) LSI-CoVA-017 (blue) are shown. Corresponding chromatograms of free antibodies LSI-CoVA-016 (green) and LSI-CoVA-017 (pink) are also indicated as controls. (e) Peak fractions indicated with arrow were analyzed by reducing SDS PAGE analysis. Supplementary Fig. 5 also shows SEC chromatogram of hexaproteins Spike + LSI-CoVA-017.

Query 9A: I thank the reviewers for their figures as reference for the reviewers. I assume that the SEC column used here was the same as in the manuscript: Superose 61535 Increase 10/300 GL gel filtration column? The profile of your Spike trimer looks not as expected- is that really your purified Spike trimer used in the experiments? In that case you have a lot of aggregates and quite a reasonable amount of Monomers.... I guess max 50% of the total "purified trimer" protein is indeed the trimer! no matter if that's fresh and not frozen Spike: The quality of this Spike protein is really BAD!!!! And I now question all the experiments performed with such material!

Response 9A: We understand the reviewer's concerns. The SEC profile shown in the above figure represents the purification profile for Spike protein (black trace). The above figure was added to

highlight the absence of Spike monomers in the sample, which was raised as a concern initially by the reviewer in the first review (Query 9). The samples were then tested for binding with antibody.

The samples from the centre peak (corresponding to trimeric Spike) were used for second-round of SEC purification to obtain homogenous Spike trimer. The SEC-MALS profile of this is shown below, which clearly indicates presence of Spike trimer, and not oligomers. Eluted fractions from the trimer peak were verified using denatured SDS polyacrylamide electrophoretic analysis, which clearly indicate the purified protein is a Spike protein! This clean Spike trimer peak sample was used for HDXMS analysis and biochemical/biophysical assays. Negative stain electron microscopy images show the homogeneity of the Spike trimers.

Many groups have shown that Spike trimers inherently may have a small fraction of monomers due to monomer-trimer equilibrium. Furthermore, we have once again repeated the SEC analysis of purified Spike from Wuhan-Hu-1 and Omicron BA.2 proteins, and the chromatograms show a single dominant peak corresponding to Spike trimer.

Figure: SEC analysis of purified (A) Wuhan-Hu-1 and (B) Omicron BA.2 Spike.

SEC chromatograms (performed on Superose 6 increase 10/300 GL analytical column) show elution of the two constructs at 13 ml elution volume, with corresponding SEC-MALS data (red spots) indicating the molar mass of the peak to be ~482 kDa that corresponding to Spike trimer purified from insect cell culture expression system.

The chromatograms provide strong *experimental evidence* of the Spike trimer as the major population, and not otherwise based on *guesses and assumptions*. In addition, HDXMS experimental analysis is sensitive to the oligomeric/monomeric nature of proteins (Calvaresi V. et.al. (Nat Commun 2023, DOI: 10.1038/s41467-023-36745-0), Costello S.M. et. al. (Nat Struct Mol Biol 2022, DOI: 10.1038/s41594-022-00735-5), Braet S.M. et. al. (eLife 2023, DOI: 10.7554/eLife.82584)). If our Spike trimer sample consisted of any aggregates, it would be reflected in the mass spectral profiles of the various peptides as broad isotopic distribution profiles. However, we did not observe any mass spectra corresponding to aggregates, as shown in figures below for Spike Wuhan-Hu-1, Delta, and Omicron BA.2 VoCs. These mass spectral data act as orthogonal approach to report the presence or absence of oligomers.

Further, the IgG antibodies using in the current study bind to structured regions of the Spike (cryptic epitopes and NTD). These epitopes are inaccessible in oligomer/aggregated Spike proteins, and our antibody-binding affinity data wouldn't show the binding profiles for aggregated species. Together,

the SEC, microscopy, and mass spectrometry results clearly indicate the samples indeed consist of Spike trimer and not aggregates, and were of high quality.

Figure: Mass Spectra of different Spike Wuhan-Hu-1 peptides. Stacked mass spectral plots of six representative peptides spanning different regions of Spike Wuhan-Hu-1 are shown. The amino acid sequence and residue numbers of the peptides are indicated. For each

peptide, isotopic mass distribution profiles (blue plots) of undeuterated (bottom), 1 min deuterium labelling (center), and 100 min deuterium labelling (top) are shown.

Figure: Mass Spectra of different peptides of Spike Delta VoC.

Stacked mass spectral plots of six representative peptides spanning different regions of Spike Delta VoC are shown. The amino acid sequence and residue numbers of the peptides are indicated.

Figure: Mass Spectra of different peptides of Spike Omicron BA.2 VoC.

Stacked mass spectral plots of six representative peptides spanning different regions of Spike Omicron BA.2 VoC are shown. The amino acid sequence and residue numbers of the peptides are indicated.

Query 10: Line 210: Why is CoVA2-04 a weak neutralizer if you yourself claimed it to be a moderate neutralizer in fig 1c?

Response 10: CoVA2-04 is a moderate neutralizer. This error in the manuscript text has been corrected.

ok.

Query 11: Line 239f This whole (and very interesting) paragraph does only have figures/tables/data in the extended data. You should think about restructuring the figures.

Response 11: We thank the reviewer for this feedback. The section and figures have been restructured in the revised manuscript to highlight the importance of these results.

The structure is better.

Query 12: Line 268: Why not test all 9 Abs on the deglycosylated Spike?

Response 12: We tested the effects of deglycosylation and glycosylation using isolated RBD and Spike constructs of Wuhan-Hu-1, Delta, and Omicron variants. This was part of the extended data in the earlier manuscript and in the revised manuscript, it is Supplementary Figure 10 with data in Supplementary Table S6.

ok.

Query 13: Line 319 “hinge dynamics” you never used this term before or afterwards so I cannot see that you explored the possibility of it?

Response 13: The term ‘hinge dynamics’ refers to the conformational changes observed at the region connecting the S1 and the S2 subunits, and has been edited in the revised manuscript. The earlier result sections reported the increased deuterium exchange at the various hotspots of the S2 subunit upon antibody binding. At this section, ‘hinge dynamics’ indicated the collective changes for the different antibodies.

is ok.

Query 14: Line 324f: “We did not observe any enhancement in the neutralization efficacies of the two potent HuMAbs (CoVA2-39, 5A6) with NTD-binding LSI-CoVA-017” probably the resolution of your method is not suitable to do so? You already detect nearly 100% using CoVA2-39 or 5A6 alone. How do you expect to see any additive effect?

Response 14: We appreciate the reviewer’s concerns regarding resolution. We would like to point out that the strongly neutralizing antibodies CoVA2-39 and 5A6 alone showed 100% neutralization at the concentrations specified, with a possibility that the addition of another antibody may change the neutralization profile. For deeper understanding of the absolute quantitative comparison, additional experiments using a wider range of varying antibody concentrations, or an alternative method may be suited. In this study, we have highlighted the qualitative nature of the antibody competition.

is ok with the addition of “This could be possibly due to the limitation of the method and warrants alternative method for quantification purposes.”

Query 15: Line 352 very interesting and conclusive result. Line 363ff: Please try to restructure this paragraph its really difficult to follow for the reader. Line375: is it surprising? Maybe its avidity effects? Fig 1b) statistical analysis, how many experiments? Fig 1c) number of experiments- standard deviation shown?

Response 15: We have revised the manuscript to ensure these details are clearly expressed and have added the statistical parameters used. Binding activity measurements and neutralization assays were

performed in triplicates, and for many points the standard deviations are too small to be visible. The raw data are tabulated as Source Data 1.

is ok.

Query 16: Figure 1d+e I don't get why this is not in one with figure 2a-c (its also described together in one section in the results part). Figure 1 d) in the legend I and ii are against RBD in the figure its I and iii whereas ii is minus NTA. Figure 1 d) in this quality numbers on x axis are not readable. Fig.2) e+f are way smaller font than a,b,c, d

Response 16: The manuscript was directly transferred from Nature, with restriction in number of figures in the main manuscript. However, we have restructured the revised manuscript for clarity. The reviewer's suggestions to merge parts of different figures were also considered. The figures and legends in the revised manuscript now have appropriate font size for reading.

The arrangement of the figures is improved.

Query 17: Fig.4) very clear figure that gives a good overview about the model.

Response 17: We thank the reviewer for their appreciation and feedback.

Query 18: Extended Data Fig.1: what does the black half brackets under diagram b mean? Legend and y axis description should be added to every figure.

Response 18: The half brackets indicated that the peptides with overlapping residue numbers were sub-grouped to prevent text overlap. In the revised figures, we have indicated the residue numbers of each peptide, which correlates with the corresponding source data (or supplementary data) as indicated in the figure legends. Both x- and y- axes have been indicated for all figures in the revised manuscript.

Query 18A: Is okay – just add also a unit to x-axis of Sup. Fig8 c

Response 18A: The units on the x-axis has been added.

Query 19: Line504: If you worked with blood samples I would expect an ethical statement?

Response 19: This work was conducted under an approved protocol from an Institutional Review Board. We have added the necessary statements to the ethical statement in the Methods section.

ok

Query 20: Line 518/519 what kind of branched Pei (e.g. Mw 600 or 70000), what company?

Response 20: We used branched PEI stock solution (Sigma Cat. No. 408727) with an average Mw of ~25,000 by LS, and average Mw ~10,000 by GPC.

ok.

Query 21: Line 514f - In general the transfection protocol is missing information like cell density, media, transfection volume, PEI : DNA ratio.

Response 21: The revised manuscript describes the methods in significantly greater detail. Briefly, mammalian cells were cultured in freestyle f17 expression medium (Life Technologies). Cultures at a

density of 1×10^6 cells per mL were transfected with 1mg of DNA per mL of culture at 1:3 ratio of DNA: branched PEI (Sigma-Aldrich). The culture supernatant was harvested for purification at day 6 post transfection.

ok.

Query 22: Line 514f - As the glycosylation differs in Sf9 cells and HEK cells you should specify in the experiments if you used insect cell or HEK cell derived material.

Response 22: In the revised manuscript, we have specified the nature of Spike constructs used for different experiments. Spike trimer purified from insect-cell culture was mainly used for HDX analysis, glycosylated vs deglycosylated Spike binding affinity to antibodies by QCM, while Spike HexaPro trimer purified from mammalian-cell culture was used for other binding analyses.

Query 22A: It is still not clear to me in the revised manuscript where you used what material where did you put that information? – e.g. Figure 1 was it insect cell derived or mammalian ? - I only found the term “insect” 2x in the manuscript... –You should at least add this information in the material section or in the description of each figure.

Response 22A: The expression and purification of Spike protein Wuhan-Hu-1 and its VoCs were mentioned in the materials section. We have added the sources of Spike protein in the description of each figures.

Query 23: Line 514ff: what about recombinant ACE2 -didn't you use that as well, how was it produced?

Response 23: Expression and purification of recombinant ACE2 was described in our previous manuscript – which has been cited: Raghuvamsi, P. V. et al. SARS-CoV-2 S protein:ACE2 interaction reveals novel allosteric targets. eLife 10 (2021).

Query 23A: Please mention this in the manuscript and cite the paper e.g. in the method section for ACE2 production.

Response 23A: Our reference corresponding to the purification of ACE2 is cited in the methods section (under Antibody and antigen expression and purification).

Query 24: Line 539 please add information about Hexa Pro spike variant e.g. no furin site as the reader does have to look it up on addgene. Again it is unclear in what experiments you used what variant and why you chose the Hexapro variant.

Response 24: We thank the reviewer for highlighting this point. The revised manuscript has been modified to prevent any inconvenience or misunderstanding of the nature of the Spike construct used. This information regarding the Spike protein derived from the Wuhan-Hu-1 strain of virus was described in detail in our previous manuscript and cited accordingly. The details of the Spike constructs and the mutations have been added in the Methods section of the revised manuscript.

ok.

Query 25: Line 981 RBD (Wuhan)

Response 25: We have amended this in the revised manuscript.

ok

Query 26: Line 1014: I cannot find ACE2-binding inhibition assay in material and methods?

Response 26: We thank the reviewer for pointing this out. The details for the ACE2-binding inhibition assay are now described in significant detail in the Methods section of the revised manuscript.

ok

Query 27: Line 1095 should be Table S3

Response 27: We have amended this in the revised manuscript.

ok.

Reviewers' Comments:

Reviewer #1:

Remarks to the Author:

The authors clarified all raised points to my satisfaction